# The calcium pump PMCA4b promotes epithelial cell polarization and lumen formation
Sarolta Tóth [1] ✉, Diána Kaszás[1,2,3], János Sónyák[1], Anna-Mária Tőkés[4], Rita Padányi[4,5], Béla Papp[6,7,8], Réka Nagy [1,3], Kinga Vörös[3,4,9], Tamás Csizmadia[10], Attila Tordai[1] & Ágnes Enyedi [1,11] ✉

Loss of epithelial cell polarity and tissue disorganization are hallmarks of carcinogenesis, in which $Ca^{2+}$ signaling plays a significant role. Here we demonstrate that the plasma membrane $Ca^{2+}$ pump PMCA4 (ATP2B4) is downregulated in luminal breast cancer, and this is associated with shorter relapse-free survival in patients with luminal A and B1 subtype tumors. Using the MCF-7 breast cancer cell model we show that PMCA4 silencing results in the loss of cell polarity while a forced increase in PMCA4b expression induces cell polarization and promotes lumen formation. We identify Arf6 as a regulator of PMCA4b endocytic recycling essential for PMCA4-mediated lumen formation. Silencing of the single *pmca* gene in *Drosophila melanogaster* larval salivary gland destroys lumen morphology suggesting a conserved role of PMCAs in lumen morphogenesis. Our findings point to a role of PMCA4 in controlling epithelial cell polarity, and in the maintenance of normal glandular tissue architecture.

Breast cancer is the most commonly occurring cancer in women, representing about 10% of all cancers in the female population worldwide[1]. The majority of breast cancers are estrogen (ER+) and progesterone (PgR+) hormone receptor-positive, of which the luminal A (LUMA) subtype is the least aggressive (low grade) while the more proliferative (Ki-67-high) luminal B1 (LUMB1) and human epidermal growth factor receptor 2-positive (HER2+) luminal B2 (LUMB2) subtypes are usually of higher grades.

Intracellular $Ca^{2+}$ homeostasis is frequently altered in cancer cells, and this can play a role in tumor initiation, progression, angiogenesis and metastasis[2,3]. Plasma membrane calcium ATPases (PMCA/ATP2B) regulate intracellular $Ca^{2+}$ concentration by pumping $Ca^{2+}$ out of the cytoplasm into the extracellular space in order to maintain the exceedingly low cytosolic $Ca^{2+}$ concentration. In mammalian cells four genes (*ATP2B1-4*) encode PMCA1-4 proteins, and as a result of alternative splicing, more than 20 PMCA isoforms have been described. PMCA1 is the house-keeping form that together with PMCA4 is expressed in all cell types while PMCA2 and PMCA3 expressions are tissue specific[4,5]. Both PMCA2 and PMCA4 play a

role in the adult mammary gland, but their functions are quite different[6]. PMCA2 is highly expressed during the lactating period and is responsible for setting $Ca^{2+}$ concentration in the milk[7]. In contrast, PMCA4 is upregulated before lactation and during mammary gland involution[6,8]. It has been well established that serum estradiol levels gradually increase during pregnancy, and we showed that the "b" splice variant of PMCA4 (PMCA4b) expression was highly dependent on estradiol concentrations in ER-α positive MCF-7 breast cancer cells, suggesting a role for this pump in mammary gland development[9].

Alterations in PMCA expression have been reported in various cancer types[3,10]. Among these, PMCA4b was shown to be downregulated in ER+ luminal-type breast cancer cells whereas its expression was relatively high in basal-type cells with, however, a predominant localization to intracellular compartments[9]. Low expression of PMCA4b was detected in highly metastatic BRAF mutant melanoma cell lines, in which re-expression of PMCA4b was associated with decreased migration and metastatic activities suggesting a metastasis suppressor role for this PMCA variant[11]. In contrast, PMCA2 mRNA expression was upregulated in HER2-positive and

[1]Department of Transfusion Medicine, Semmelweis University, Budapest, Hungary. [2]Department of Physiology, Semmelweis University, Budapest, Hungary. [3]School of PhD Studies, Semmelweis University, Budapest, Hungary. [4]Department of Pathology, Forensic and Insurance Medicine, Semmelweis University, Budapest, Hungary. [5]Department of Biophysics and Radiation Biology, Semmelweis University, Budapest, Hungary. [6]Institut National de la Santé et de la Recherche Médicale, Inserm UMR 1342, Institut de Recherche Saint-Louis, Hôpital Saint-Louis, Paris, France. [7]Institut de Recherche Saint-Louis, Hôpital Saint-Louis, Université de Paris, Paris, France. [8]CEA, DRF-Institut Francois Jacob, Department of Hemato-Immunology Research, Hôpital Saint-Louis, Paris, France. [9]Institute of Translational Medicine, Semmelweis University, Budapest, Hungary. [10]Department of Anatomy, Cell and Developmental Biology, Eötvös Loránd University, Budapest, Hungary. [11]ELKH-SE Biophysical Virology Research Group, Eötvös Loránd Research Network, Budapest, Hungary. ✉e-mail: sarolta.toth7@gmail.com; enyedi.agnes@semmelweis.hu

basal-type breast cancers[12], and its expression correlated with HER2 levels in HER2-prositive tumors[13].

Luminal epithelial cells display apical-basal polarity that is regulated by specific protein modules under the control of the Par, Crumbs and Scribble polarity protein complexes[14]. Lumen formation is an essential step in mammary gland development, and loss of lumens is a hallmark of tumorigenesis[15]. It has been shown that primary mammary epithelial cells isolated from mice were able to form lumen-containing ducts on reconstituted basement membrane matrix, however, after oncogene induction, the cells lost their epithelial polarity and lumen-forming capability[16]. In polarized cells directed vesicular transport is essential to ensure proper protein and lipid composition of specialized membrane compartments[17]. The ADP-ribosylation factor 6 (Arf6) small GTPase protein regulates membrane trafficking pathways[18], and it has been implicated in cell polarization[19–21] and lumenogenesis[22,23]. Although Arf6 is essential for cell polarization, Arf6 hyperactivity can promote invasiveness and metastatic activity of cancer cells[18,24] by increasing their motility[25].

In the present study we show that PMCA4(b) (assessed by a PMCA4-specific antibody) is downregulated in luminal-type breast carcinoma tissue samples, and that the expression level of the *ATP2B4* gene correlates with patient survival. We demonstrate that PMCA4b promotes polarization and lumen formation of ER$^+$ MCF-7 breast cancer cells, and this requires the previously identified di-leucine endocytic motif at the C-terminus of the pump[26]. Here we find that PMCA4b shows strong colocalization with wild-type and constitutively active forms of Arf6, moreover inhibition of Arf6 leads to the intracellular accumulation of PMCA4b that defines Arf6 as a regulator of PMCA4b trafficking. To investigate the role of PMCAs in lumen formation we silenced the single *Drosophila melanogaster* PMCA (dmPMCA) in larval salivary gland, and this disrupted lumen morphology and induced a severe secretion defect suggesting an evolutionarily conserved role of PMCAs in lumen formation and secretion.

## Results

### PMCA4 is downregulated in HR$^+$ breast cancer

Besides an analysis of publicly available gene expression datasets[9], no clinically relevant data have been published on PMCA4 expression in breast cancer. Since a majority of all breast cancer cases (about 70–80%) are hormone receptor-positive (HR$^+$)[27], we investigated PMCA4 protein abundance in HR$^+$ breast carcinoma tissue samples (Supplementary Data) using the PMCA4-specific antibody JA9 (JA9 does not discriminate between PMCA4 "a" and "b" splice variants)[28], and compared the results to those of normal breast tissue samples obtained after breast reduction surgery. We found that PMCA4 was highly expressed in the ductal epithelium of normal breast tissue and in more differentiated regions of tumor samples when compared to fully de-differentiated regions of the tumor (Fig. 1a). In 76% of the breast cancer cases (83 out of 109 tissue samples) less than 5% of the tumor cells showed PMCA4 staining. Analyzing separately the HR$^+$ subtypes, no statistically significant differences were found between the LUMA and LUMB cases (Fig. 1b and Supplementary Data). Data from the Clinical Proteomic Tumor Analysis Consortium[29] (CPTAC) (https://ualcan.path.uab.edu) confirmed these findings and showed significantly reduced PMCA4 both at the mRNA (The Cancer Genome Atlas, TCGA) and protein levels in luminal breast cancer types compared to normal breast tissue. In contrast to PMCA4, PMCA1 expression was reduced at the mRNA level while no significant difference could be detected at the protein level in the same comparison (Fig. 1c and Supplementary Fig. 1a). Importantly, the mRNA level of PMCA2 was much lower in normal non-lactating breast tissue compared to the other isoforms, and no significant difference was observed between normal tissue and luminal breast cancer cohorts (Supplementary Fig. 1a).

To assess the prognostic significance of ATP2B4 (PMCA4) expression in the clinical outcome of HR$^+$ breast carcinoma subtypes, the publicly available KM Plotter online tool[30] (www.kmplot.com) was used. According to this database, high ATP2B4 gene expression was associated with greater probability of relapse-free survival (RFS) in grade 2 LUMA and in LUMB1

tumors, whereas an inverse correlation was observed in the HER2-expressing LUMB2 subtype, in which poorer outcomes were associated with higher ATP2B4 expression. It is worth noting that in grade 1 and 3 LUMA tumors the effects were not significant (Supplementary Fig. 1b). In contrast to ATP2B4, no statistically significant association was detected between low and high ATP2B1 gene expression and RFS in the LUMA and LUMB1 breast carcinoma subtypes, while in the LUMB2 subtype high ATP2B1 expression (PMCA1) level was associated with a lower probability of RFS as also seen for high ATP2B4 (PMCA4) expression (Fig. 1d).

### PMCA4b silencing induced internalization of E-cadherin in MCF-7 breast cancer cells

High PMCA4 protein levels in the normal mammary gland and its nearly complete loss in luminal subtype tumors suggest that PMCA4 is involved in the development and/or maintenance of the normal mammary tissue. The PMCA4b splice variant is a ubiquitous form of PMCA4, and this isoform has been found in several luminal-type breast cancer cells including the estrogen-sensitive MCF-7 cell line[9,31]. Although these cells express low levels of PMCA4b, they retain most of their epithelial features since they express the epithelial marker E-cadherin but not the mesenchymal markers N-cadherin and vimentin[32]. To understand the role of PMCA4(b) loss during tumor progression, the endogenous PMCA4b was silenced by a PMCA4-specific short hairpin RNA (sh-RNA), or the GFP-tagged PMCA4b and its endocytosis-defective mutant form PMCA4b$^{LA}$[26] were stably over-expressed in MCF-7 cells. We found that neither silencing nor over-expression of PMCA4b affected the expression levels of the epithelial and mesenchymal markers E-cadherin, N-cadherin and vimentin (Fig. 2a and Supplementary Fig. 2a, b), respectively, or influenced cell growth (Supplementary Fig. 2c). Although silencing PMCA4 did not affect significantly the overall expression level of E-cadherin (Supplementary Fig. 2b), it did result in its internalization and localization in intracellular compartments (Fig. 2b–e). In contrast, the parental, GFP-PMCA4b- or the trafficking-mutant GFP-PMCA4b$^{LA}$-expressing MCF-7 cells displayed more prominent plasma membrane localization of E-cadherin (Fig. 2b–e and Supplementary Fig. 2d–f). This suggests that loss of PMCA4 may induce partial epithelial-mesenchymal transition (EMT) through E-cadherin internalization, without significantly affecting the overall expression of E-cadherin and other EMT markers, similarly to published data[33].

### PMCA4b regulates epithelial cell polarity

E-cadherin is involved in cell-cell adhesion and cell polarization partly through its interaction with scribble (Scrib), a member of the apico-basal Scribble polarity complex that consists of lethal giant larvae (Lgl), discs large 1 (Dlg1) and Scrib proteins[34]. It has been known for a long time that the C-terminal PDZ-binding sequence motif ETSV of PMCA4b binds to the PDZ-domains 1 and 2 of the scaffold protein Dlg1[35] (Fig. 2f), and this way it can participate in cell polarization. Here we show that Dlg1 is highly expressed in MCF-7 cells and its expression is not affected by either PMCA4-silencing or PMCA4b overexpression (Fig. 2a). Fluorescence immunostaining of PMCA4b-expressing MCF-7 cells show colocalization between PMCA4b and Dlg1 at the plasma membrane, as expected (Fig. 2g), while PMCA4-silencing is associated with preferential cytoplasmic Dlg1-localization compared to wild type PMCA4b-overexpressing cells (Fig. 2h–k and Supplementary Fig. 2g), suggesting that PMCA4b is involved in targeting Dlg1 to the plasma membrane.

Ezrin is an F-actin-binding protein that is known to localize to the apical plasma membrane of polarized epithelial cells[36]. The overall high level and/or atypical localization of ezrin in tumor cells were shown to correlate with poor prognosis in breast cancer[37,38]. While expression of sh-PMCA4, PMCA4b or the trafficking mutant PMCA4b$^{LA}$ in MCF-7 cells did not alter ezrin protein levels (Fig. 3a), variations in PMCA4b expression showed strikingly different ezrin localization patterns. In contrast to the PMCA4-silenced or trafficking mutant PMCA4b$^{LA}$-expressing MCF-7 cells where a substantial proportion of ezrin was

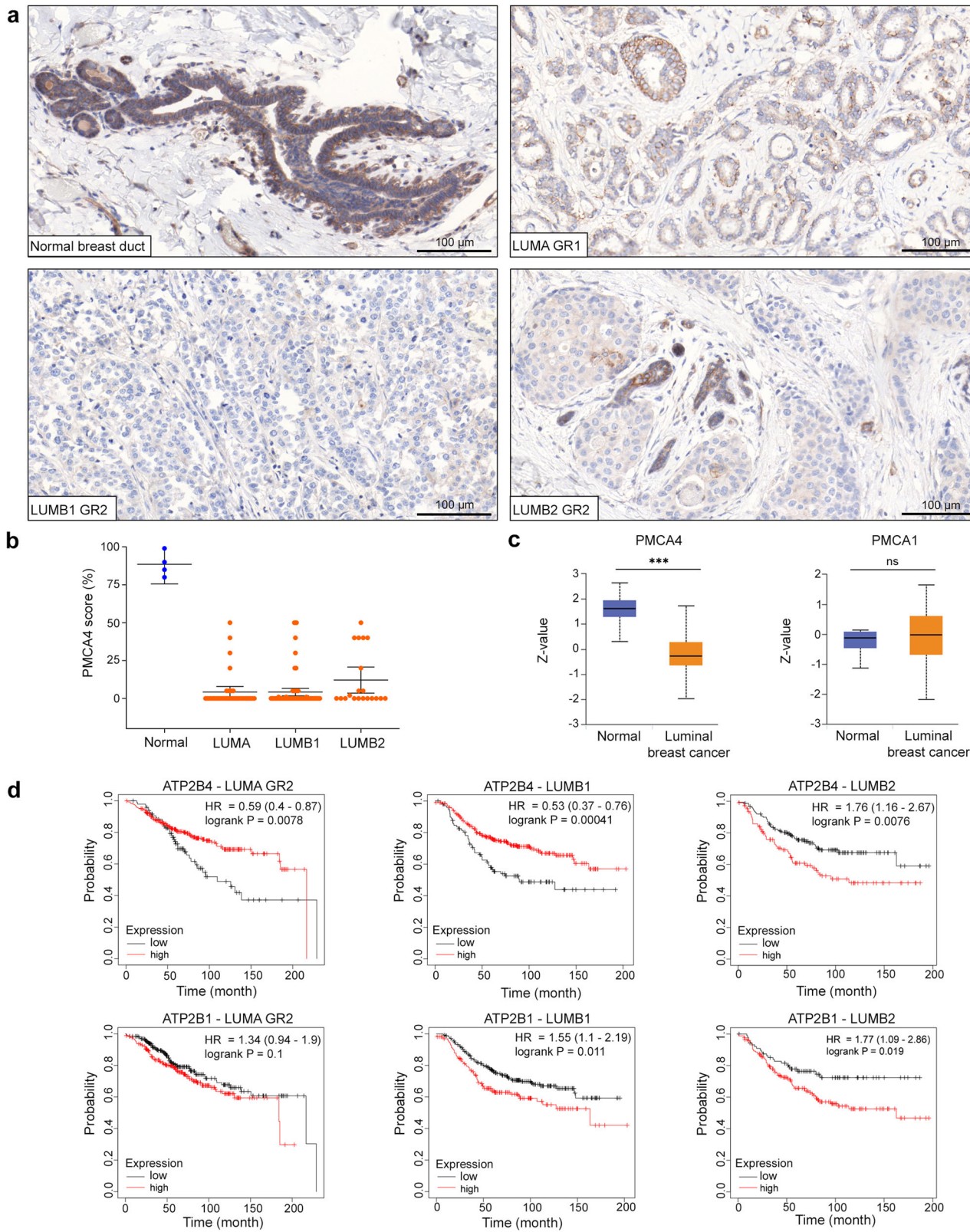

localized to intracellular compartments, PMCA4b overexpression resulted in a significantly more prominent asymmetric ezrin localization when compared to the parental MCF-7 cells (Fig. 3b–d). These data suggest that PMCA4b abundance affects cell polarization and show that not only PMCA4b expression but also its proper trafficking is required to accomplish cell polarity.

## PMCA4b is a critical mediator of Ca²⁺ signaling

Prolonged Ca²⁺ signaling has been shown to induce partial EMT in epithelial cells, characterized by the internalization of E-cadherin[39]. Previous studies suggest that in breast epithelial cells, PMCA4 plays a critical role in removing Ca²⁺ after stimuli to terminate the Ca²⁺ signal[5,31]. Here, we demonstrate that purinergic stimuli induced by extracellular ATP resulted

**Fig. 1 | PMCA4 is downregulated in HR⁺ breast cancer subtypes. a** PMCA4-specific antibody staining of normal and neoplastic LUMA, LUMB1 and LUMB2 subtype mammary gland tissue samples. GR1 or 2 indicates histological grade 1 or grade 2. **b** Fraction of cancer cells showing positive membranous PMCA4 staining; $n_{(normal)} = 4$ (obtained after breast reduction surgery), $n_{(LUMA)} = 39$, $n_{(LUMB1)} = 50$, $n_{(LUMB2)} = 20$ tissue samples. Error bars show 95% confidence intervals (CI). **c** Protein expression of PMCA4 (NP_001675.3:S328) and PMCA1 (NP_001673.2:S1182) in normal breast tissue and luminal breast cancer subtypes. Data derived from the Clinical Proteomic Tumor Analysis Consortium (CPTAC) (http://ualcan.path.uab.edu) that are expressed as $z$-values and analyzed by Student's $t$ test; $n_{(PMCA4b/normal)} = 18$, $n_{(PMCA4b/luminal)} = 64$, $n_{(PMCA1b/normal)} = 18$, $n_{(PMCA4b/luminal)} = 64$ tissue samples; ***$p < 0.001$, ns$p > 0.05$; ns indicates non-significant difference. Error bars show standard deviations. **d** Kaplan–Meier relapse-free survival analysis in breast cancer patients with LUMA, LUMB1 or LUMB2 subtype tumors with low and high *ATP2B4*/PMCA4 and *ATP2B1*/PMCA1 expression levels. GR 2 indicates histological grade 2. Data were collected by the online survival analysis tool (http://www.kmplot.com) using microarray data analysis; $n_{(ATP2B4/LUMA-GR2/low)} = 97$, $n_{(ATP2B4/LUMA-GR2/high)} = 297$, $n_{(ATP2B4/LUMB1/low)} = 99$, $n_{(ATP2B4/LUMB1/high)} = 301$, $n_{(ATP2B4/LUMB2/low)} = 150$, $n_{(ATP2B4/LUMB2/high)} = 108$, $n_{(ATP2B1/LUMA-GR2/low)} = 257$, $n_{(ATP2B1/LUMA-GR2/high)} = 254$, $n_{(ATP2B1/LUMB1/low)} = 239$, $n_{(ATP2B1/LUMB1/high)} = 161$, $n_{(ATP2B1/LUMB2/low)} = 89$, $n_{(ATP2B1/LUMB2/high)} = 169$ patients.

in a markedly elevated Ca²⁺ response in sh-PMCA4 and parental (PMCA4-low) cells when compared to cells overexpressing wild type PMCA4b or the trafficking mutant PMCA4b^LA (Fig. 4a, b). In cells with low PMCA4b abundance (sh-PMCA4 expressing and parental cells), the Ca²⁺ response exhibited two distinct peaks. The first peak likely corresponds to Ca²⁺ release from the endoplasmic reticulum (ER), while the second peak may involve Ca²⁺ entry through store-operated calcium channels (SOCs)[40]. In contrast, cells overexpressing PMCA4b^LA showed a rapid return to the baseline Ca²⁺ level after the first peak, with only minor fluctuations reminiscent of the second peak seen in sh-PMCA4 and parental cells in good correlation with our previous work[41]. Cells overexpressing wild type PMCA4b exhibited a significantly reduced Ca²⁺ response, as evidenced by a lower area under the curve and peak response to ATP (Fig. 4b, c). Our observations in MCF-7 cells suggest that the loss of PMCA4b leads to highly dysregulated Ca²⁺ signaling in breast cancer cells.

### PMCA4b enhances polarized vesicular trafficking
Epithelial cells require polarized vesicular trafficking for the proper distribution of proteins and lipids at specific plasma membrane domains to establish and maintain apico-basal cell polarity[42]. Wheat germ agglutinin (WGA) has been widely used to follow endosomal trafficking in live cells. Upon binding to N-acetylglucosamine and sialic acid moieties on the extracellular side of plasma membrane proteins WGA can be internalized and transported through the endosomal pathways[43]. We performed a WGA uptake assay and found that PMCA4b-expressing cells collected WGA-positive vesicles to one side of the plasma membrane (Supplementary Movie 1) as a sign of polarized vesicular trafficking while these vesicles were located randomly in the cytoplasm of the parental, the sh-PMCA4- and the PMCA4b^LA-expressing cells (Fig. 5a–d and Supplementary Movies 2 and 3). Furthermore, PMCA4b strongly co-localized with WGA-positive vesicles in contrast to the PMCA4b^LA mutant that clearly separated from the WGA-positive puncta (Fig. 5e) confirming the importance of PMCA4b trafficking in cell polarization.

### Arf6 regulates intracellular trafficking of PMCA4b
PMCAs require complex formation with neuroplastin or CD147/basigin for their proper plasma membrane localization and function[44,45]. It has been demonstrated that internalization and recycling of CD147 is mediated by Arf6[46], and Arf6 has been implicated in cell polarization[19]. To study the role of Arf6 in PMCA4b trafficking we transiently over-expressed wild type Arf6 and its constitutively active mutant form Arf6^Q67L in PMCA4b-expressing MCF-7 and HEK-293 cells. Strong co-localization between PMCA4b and wild type Arf6 was seen near the plasma membrane and intracellular compartments (Fig. 6a) while the mutant Arf6^Q67L caused intracellular accumulation of PMCA4b in both cell types (Fig. 6b), similarly to that described for other known Arf6 cargo proteins[47,48]. To inhibit Arf6 we treated PMCA4b-expressing MCF-7 cells with the Arf6 inhibitor NAV2729[49]. After 24 h of the treatment, we observed an enrichment of PMCA4b-CD147 positive vesicles near the plasma membrane, and after 48 h PMCA4b-CD147 vesicles accumulated in the cytoplasm (Fig. 6c, d, e and Supplementary Fig. 3) suggesting that Arf6 was involved in the endocytic recycling of the PMCA4b-CD147 complex. These results indicate that Arf6 is a novel regulator of PMCA4b-CD147 endosomal trafficking (Fig. 6f).

### PMCA4b promotes pre-lumen formation in 2D cultures of MCF-7 cells
The establishment and maintenance of epithelial cell polarity are hallmarks of healthy mammary tissue organization featured by a complex ductal network. However, during neoplastic transformation the polarity of neoplastic mammary epithelial cells is lost and their capability to form mature mammary ducts with well-defined central lumens is also compromised[50]. We found that in highly confluent 2D cultures of MCF-7 cells PMCA4b over-expression led to the formation of significantly more pre-lumen-like structures resembling cellular intermediates of lumen formation than in the case of the parental, the sh-PMCA4 or the trafficking mutant PMCA4b^LA-expressing cells (Fig. 7a, d, e). PMCA4b overexpression elevated the number of pre-lumen-like structures in T47D luminal breast cancer cells, as well (Supplementary Fig. 4a–c) further supporting the role of PMCA4b in lumen formation in luminal cell types. PMCA4b colocalized with phalloidin-labeled actin positive vesicles and WGA-positive endosomes near pre-lumens (Fig. 7b, c) indicating that PMCA4b participates in the dynamic vesicular trafficking characteristic of de novo lumen formation[51,52].

We treated the PMCA4b-expressing MCF-7 cells with the Arf6 inhibitor NAV2729 and found significantly fewer F-actin labeled pre-lumen structures compared to the non-treated PMCA4b-expressing cells suggesting that Arf6 activity is needed for the positive effect of PMCA4b on pre-lumen formation (Fig. 7e, f and Supplementary Fig. 5).

### PMCA4b promotes central lumen formation in MCF-7 mammospheres
To confirm the role of PMCA4b in lumen formation, we cultured MCF-7 cells in Matrigel for 10 days to create mammospheres and observed that the PMCA4b-expressing cells formed well-defined, F-actin-bordered central lumen-bearing spheroids with a single layer of cells around the lumen (Fig. 8a–c and Supplementary Fig. 6c) similar to that seen in cross-sections of healthy mammary duct[53]. While the parental MCF-7 cells also formed some central lumen-bearing mammospheres, the lumen structures were not well-defined, and their number was significantly lower compared to the cell cultures with elevated PMCA4b expression (Fig. 8a–c and Supplementary Fig. 6a). Mammospheres generated by sh-PMCA4 or PMCA4b^LA-expressing MCF-7 cells did not form central lumens; rather, they displayed several small F-actin-rich areas (Fig. 8a–c and Supplementary Fig. 6b, d) resembling the microlumens commonly seen in ductal carcinoma in situ (DCIS)[54]. These data indicate that both PMCA4b abundance and proper localization were required for appropriately positioned central lumen formation.

Electron microscopy revealed enrichment of intracellular vesicles in proximity to the plasma membrane. However, in contrast to the parental and PMCA4b-expressing MCF-7 cells, these vesicles—especially the dense core-types—showed excessive accumulation in sh-PMCA4-expressing MCF-7 cells, indicating a secretion defect in the absence of PMCA4b (Fig. 8d). Moreover, in PMCA4b^LA-expressing cells small endocytic vesicles accumulated near the plasma membrane (Fig. 8d). These findings further support the importance of PMCA4b as an essential regulator of vesicular trafficking and its possible role in secretion.

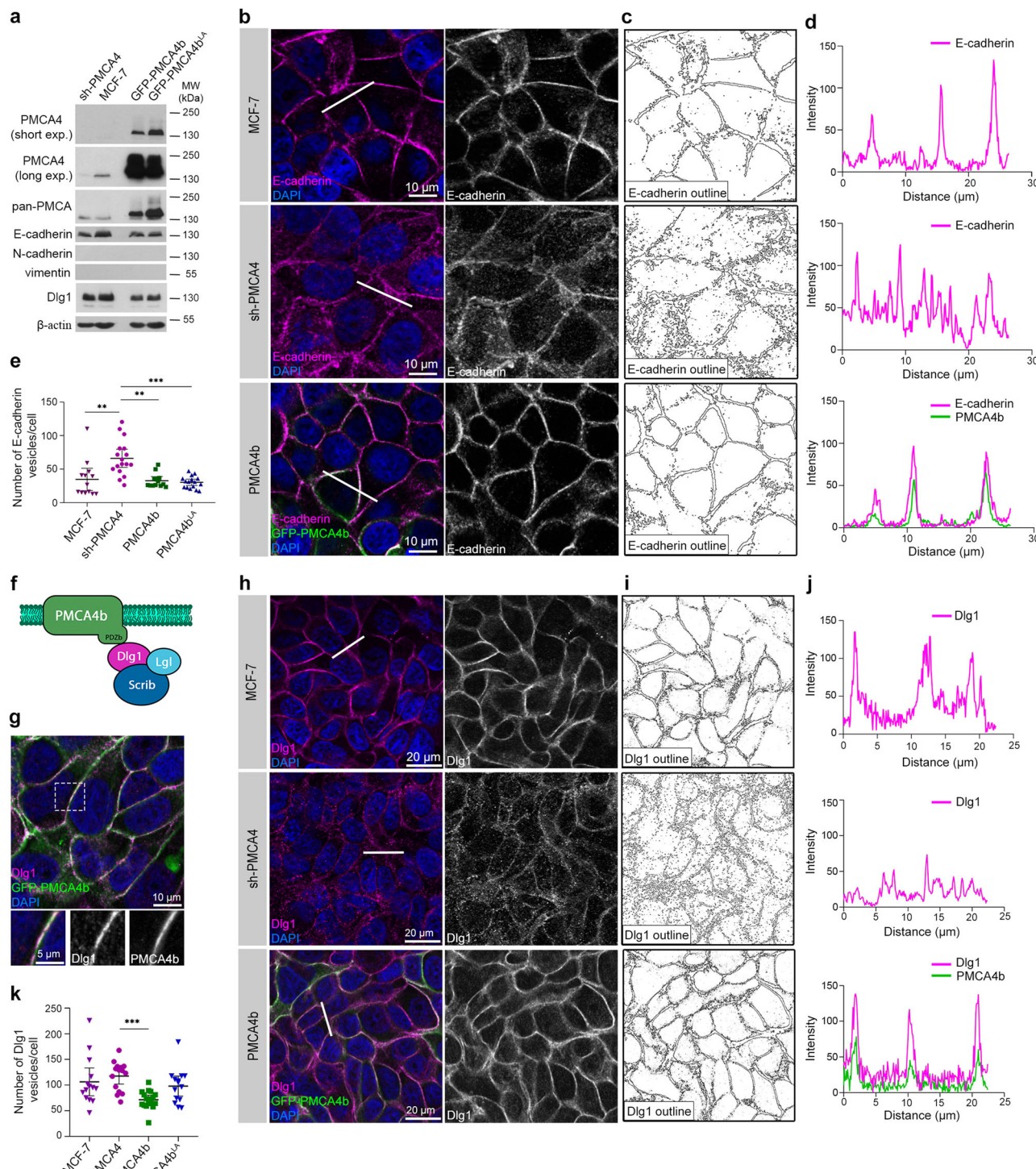

## Drosophila PMCA regulates normal lumen morphology in larval salivary gland

The only *D. melanogaster* (dm)PMCA (UniProt: Q9V4C7) shows 62.7% identity in amino acid sequence with the human PMCA4b protein (Uni-Prot: P23634-6) (Supplementary Fig. 7a). For testing the expression and localization of dmPMCA in the larval salivary gland we used the *pan*-PMCA antibody 5F10 and showed that it was suitable for the detection of dmPMCA because of its highly conserved epitope (Fig. 9a, c and Supplementary Fig. 7a). Endogenous dmPMCA localized to the basolateral and apical plasma membrane of the salivary gland cells (Fig. 9a) and showed strong co-localization with the *D. melanogaster* Dlg protein (Fig. 9b) in good

agreement with the potential PDZ-binding sequence present at the C-terminus of six out of the seven dmPMCA isoforms (Supplementary Fig. 7b).

To further study the role of PMCA in glandular development, we silenced dmPMCA using a dmPMCA-specific *UAS* promoter-driven RNA interference construct in the UAS-GAL4 system with *daughterless* promoter-driven Gal4 that effectively reduced dmPMCA expression in 3rd stage larvae (Fig. 9c). Tissue-specific silencing of dmPMCA resulted in an aberrant F-actin pattern that normally borders the central lumen of the salivary gland (Fig. 9d). In addition, cross-sections of epoxy resin-embedded salivary gland samples showed a substantial reduction in

**Fig. 2 | PMCA4b is necessary for the plasma membrane localization of E-cadherin and Dlg1. a** Western blot analysis of parental, PMCA4 specific shRNA (sh-PMCA4), GFP-PMCA4b and GFP-PMCA4b$^{LA}$-expressing MCF-7 cell lines; exp. denotes time of exposure. **b** E-cadherin immunostaining of parental, PMCA4-specific shRNA (sh-PMCA4) and GFP-PMCA4b-expressing MCF-7 cell lines. DAPI (4′, 6-diamidino-2-phenylindole, blue) labels nuclei. **c** Outline of the E-cadherin positive cell compartments on B generated by the ImageJ software. **d** Fluorescence intensity profiles of E-cadherin and GFP-PMCA4b corresponding to the white line across the cells shown in (**b**). **e** Statistical analysis of the number of E-cadherin positive vesicles in parental, PMCA4 specific shRNA (sh-PMCA4), GFP-PMCA4b and GFP-PMCA4b$^{LA}$-expressing MCF-7 cells. Graph displays the means of the number of intracellular vesicles per cell with 95% confidence intervals; data were collected from 2 independent experiments, $n_{(MCF-7)} = 13$, $n_{(sh-PMCA4)} = 17$, $n_{(PMCA4b)} = 13$, $n_{(PMCA4b^{LA})} = 16$ images. Data were analyzed with Kruskal–Wallis and Dunn's multiple comparisons tests, adjusted $p$ values: **$p < 0.01$, ***$p < 0.001$. Non-significant differences are not indicated. **f** Model of the interaction between

PMCA4b and the Scrib-Dlg1-Lgl polarity complex. PDZb denotes the PDZ-binding sequence motif of PMCA4b. **g** Dlg1 immunostaining of GFP-PMCA4b-expressing MCF-7 cells. Insets show magnification of the area framed by the dotted line. DAPI labels nuclei. **h** Dlg1 immunostaining of parental, PMCA4 specific shRNA (sh-PMCA4) and GFP-PMCA4b-expressing MCF-7 cells. DAPI labels nuclei. **i** Outline of the Dlg1 positive cell compartments on (**h**) generated by the ImageJ software. **j** Fluorescence intensity profiles of Dlg1 and GFP-PMCA4b corresponding to the white line across the cells shown on (**h**). **k** Statistical analysis of the number of Dlg1 positive particles in parental, PMCA4-specific shRNA (sh-PMCA4) and GFP-PMCA4b-expressing MCF-7 cells. Graph displays the means of the number of intracellular vesicles per cell with 95% confidence intervals; data were collected from 3 independent experiments, $n_{(MCF-7)} = 14$, $n_{(sh-PMCA4)} = 15$, $n_{(PMCA4b)} = 17$, $n_{(PMCA4bLA)} = 14$ images; data were analyzed with Kruskal–Wallis and Dunn's multiple comparisons tests; adjusted $p$ values: ***$p < 0.001$; non-significant difference is not labeled.

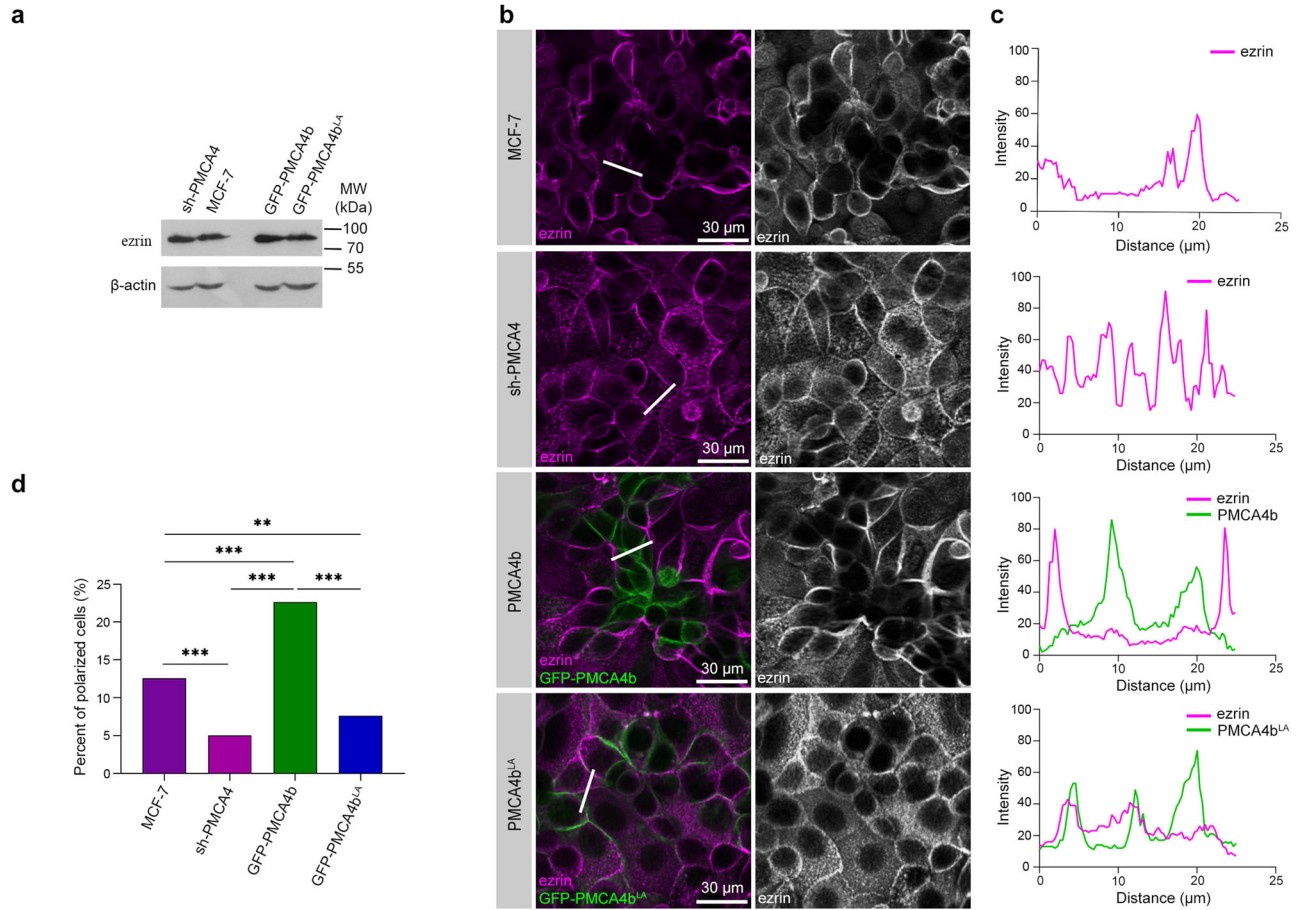

**Fig. 3 | PMCA4b induces polarized ezrin distribution. a** Western blot analysis for ezrin in parental, PMCA4-specific shRNA (sh-PMCA4), GFP-PMCA4b and GFP-PMCA4b$^{LA}$-expressing MCF-7 cell lines. **b** Ezrin immunostaining of parental, PMCA4-specific shRNA (sh-PMCA4), GFP-PMCA4b and GFP-PMCA4b$^{LA}$-expressing MCF-7 cell lines. DAPI labels nuclei. **c** Fluorescence intensity profiles of ezrin and GFP-PMCA4b corresponding to the white lines shown on (**b**). **d** Statistical

analysis of cell polarization rate based on ezrin distribution in parental, PMCA4-specific shRNA (sh-PMCA4), GFP-PMCA4b and GFP-PMCA4b$^{LA}$-expressing MCF-7 cell cultures. Data were collected from 16 fields of view from 2 independent experiments; $n_{(MCF-7)} = 1589$, $n_{(sh-PMCA4)} = 1699$, $n_{(PMCA4b)} = 1462$, $n_{(PMCA4b^{LA})} = 1511$ cells. Data were analyzed with chi-square test, $p$ values: **$p < 0.01$, ***$p < 0.001$; non-significant difference is not labeled.

central lumen size upon dmPMCA silencing (Fig. 9e, f). Closer inspection of the dmPMCA-deficient salivary gland cells by electron microscopy revealed an extensive secretory vesicle accumulation in the cytoplasm (Fig. 9G), suggesting a pronounced secretion defect. These results highlight the evolutionary conserved role of PMCA in the regulation of lumen morphology, and more specifically merocrine secretion in the *Drosophila* larval salivary gland.

## Discussion

Altered histological differentiation in malignancies (including breast cancer) characterized by the loss of tissue architecture and loss of cell polarity is associated with poor patient survival[55]. Normal mammary gland luminal epithelial cells exhibit apico-basal polarity, and this is progressively lost during tumorigenesis[15]. In the current report, we show for the first time that PMCA4 is downregulated in tumor tissue samples collected from patients

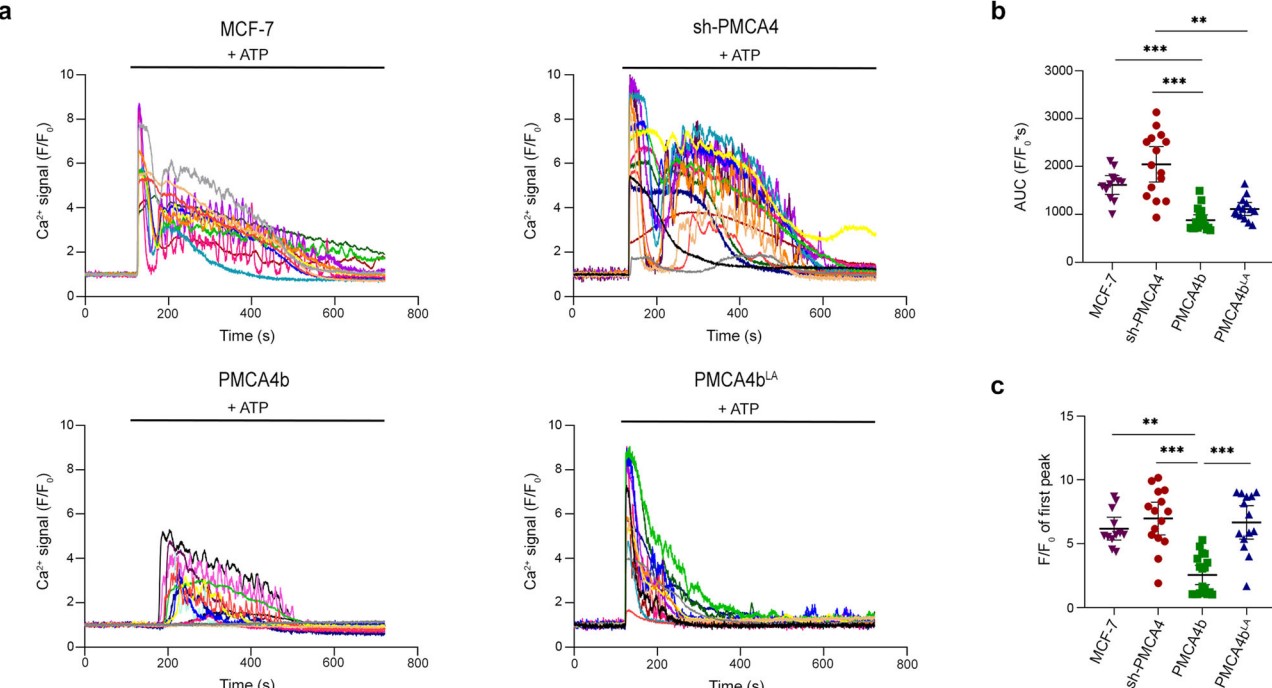

**Fig. 4 | PMCA4b mediates purinergic receptor-activated Ca$^{2+}$ signaling in MCF-7 cells. a** Intracellular Ca$^{2+}$ was measured with the R-GECO fluorescence indicator in parental, PMCA4 specific shRNA (sh-PMCA4), GFP-PMCA4b and GFP-PMCA4b$^{LA}$-expressing MCF-7 cells. 10 μM ATP was added at the 120 s mark. Signals from individual cells are illustrated using different colored lines. Number of analyzed cells: $n_{(MCF-7)} = 12$, $n_{(sh-PMCA4)} = 15$, $n_{(PMCA4b)} = 18$, $n_{(PMCA4b^{LA})} = 14$. Data are derived from 2 independent experiments. **b** Area under curve (AUC) values

were calculated from curves in panel (**a**) and analyzed with Kruskal–Wallis and Dunn's multiple comparisons tests; adjusted $p$ values: **$p < 0.01$, ***$p < 0.001$; non-significant difference is not labeled. Graph displays means with 95% confidence intervals. **c** First peak F/F$_0$ maximum values were calculated from data in (**a**) and analyzed with Kruskal–Wallis and Dunn's multiple comparisons tests; adjusted $p$ values: **$p < 0.01$, ***$p < 0.001$; non-significant difference is not labeled. Error bars show 95% confidence intervals.

with luminal breast cancer when compared to healthy breast tissue, where PMCA4 expression is high. These observations are in good correlation with CPTAC protein and mRNA expression levels of luminal breast cancer cases. In addition, we also demonstrate that elevated PMCA4 expression is associated with longer relapse-free survival in LUMA and LUMB1 breast cancer subtypes (Fig. 1). It is important to note that in contrast to the elevated levels of PMCA2 reported in HER2-positive and basal-type tumors[12,13], PMCA2 expression remains low in luminal breast tumor subtypes according to data from the CPTAC-GWAS databases (Supplementary Fig. 1a). To study whether the severe loss of PMCA4b in luminal breast cancer cells can be related to the loss of cell polarity we either silenced or increased the level of PMCA4b in MCF-7 cells. Silencing of PMCA4b caused cytoplasmic localization of E-cadherin while it did not affect its overall expression, nor did it influence the expression of mesenchymal markers - vimentin and N-cadherin—or cell growth (Fig. 2) suggesting partial EMT.

Partial EMT is a metastable hybrid state of tumor cells that exists between fully epithelial and fully mesenchymal characteristics, representing a distinct subpopulation within the tumor[56]. Recent findings suggest that partial EMT can involve the internalization of epithelial proteins, such as E-cadherin, rather than their transcriptional suppression[33]. This transient relocalization of E-cadherin in endocytic vesicles allows for the dynamic modulation of cell-cell adhesion, contributing to tumor cell invasiveness by enabling the re-establishment of cell-cell contacts during collective cell migration. Interestingly, prolonged Ca$^{2+}$ signaling has been implicated in partial EMT through its role in E-cadherin internalization[39]. Notably, low levels of PMCA4 protein, observed in various tumor types (including luminal type breast cancer cells and tissues) are associated with prolonged Ca$^{2+}$ signaling[5,31,57], and may contribute to partial EMT.

Enhanced Ca$^{2+}$ signaling was recently observed in organoids derived from breast cancer tissues in response to purinergic stimuli[58], and this

mirrors the patterns seen in MCF-7 cells with low-PMCA4 abundance in our work. In contrast, the diminished Ca$^{2+}$ signaling in MCF-7 cells with high PMCA4b abundance resembles that of organoids from normal tissue. This further supports the notion that PMCA4b can prevent sustained Ca$^{2+}$ signals, which may otherwise promote EMT and contribute to the maintenance of the malignant phenotype. Indeed, our findings demonstrate that PMCA4b is essential for the plasma membrane localization of E-cadherin and Dlg1 in MCF-7 cells (Figs. 2 and 3). Elevated PMCA4b levels also increase the number of polarized cells and promote polarized endosomal vesicular trafficking (Figs. 3 and 4) These results suggest that PMCA4b enhances cell polarization, possibly through its interaction with the PDZ polarity protein Dlg1 (Figs. 2 and 9). Interestingly, the loss of Dlg1 has been implicated in cancer progression across various tumor types highlighting the importance of its proper expression and/or localization[59–61]. Notably, in contrast to our findings in MCF-7 cells, PMCA4 silencing in gastric cancer cells induces full EMT, suggesting cancer tissue-specific function of PMCA4[62].

Loss of epithelial cell polarity is a hallmark of progression to DCIS, in which malignant cells lack the ability to form normal ductal structures, but they still express the epithelial marker E-cadherin[50]. Here we show that PMCA4b over-expressing MCF-7 cells can create mammospheres with distinct central lumens surrounded by a single layer of cells, whereas PMCA4-silenced and PMCA4b$^{LA}$-expressing MCF-7 cells formed micro-lumens typically found in DCIS (Fig. 8)[50]. Higher capability of PMCA4b-expressing cells to form a central lumen is also supported by the significantly elevated proportions of pre-lumen-like structures in 2D cell cultures compared to the parental or PMCA4-silenced MCF-7 cells (Fig. 7). In a previous paper, the authors reported that MCF-7 and T47D luminal A-type breast cancer cells stably expressing Int-αvβ3 differentiated into well-organized "acinar-like" structures both in vitro and in vivo[63], resembling the central lumen-forming mammospheres generated by our PMCA4b-expressing

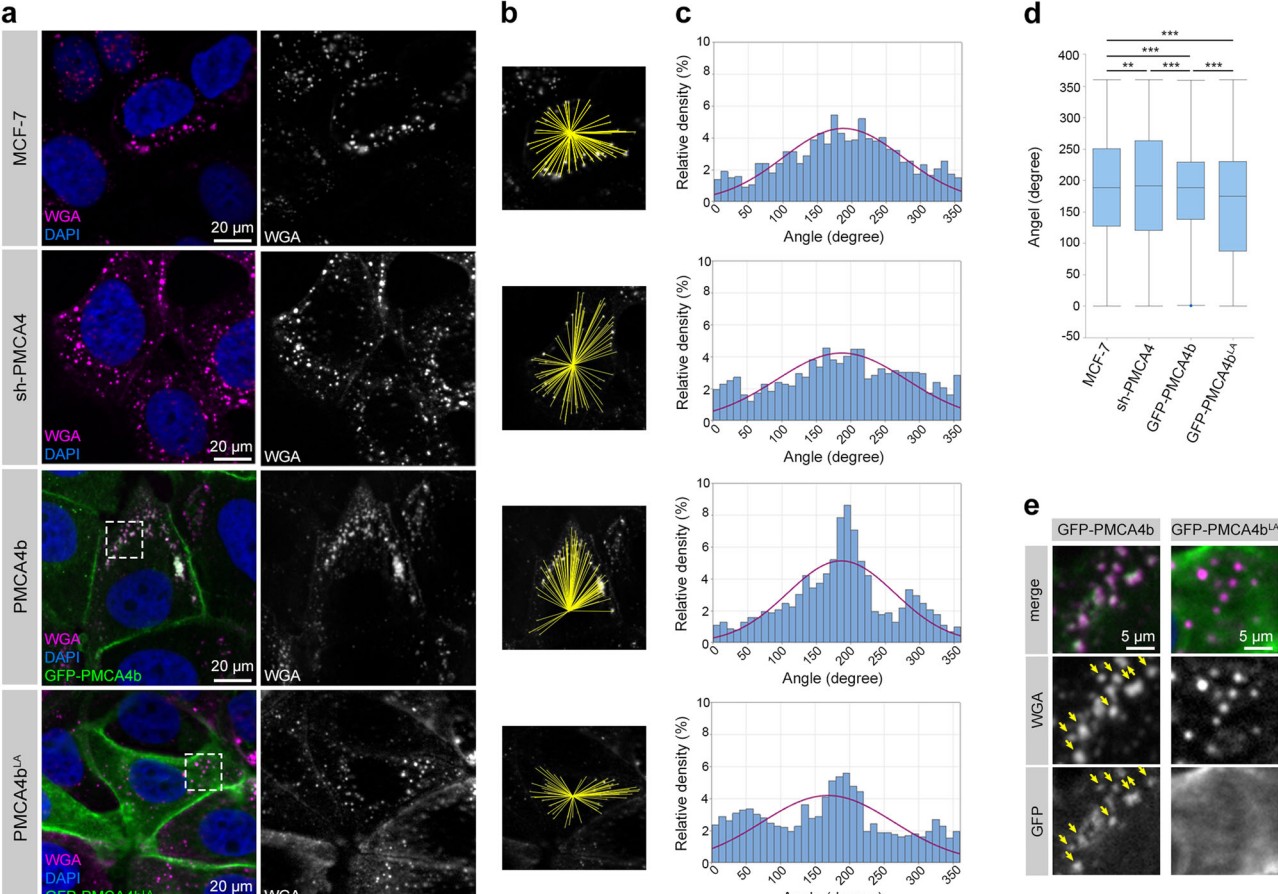

**Fig. 5 | PMCA4b promotes polarized vesicular trafficking. a** WGA uptake assay was performed on parental, PMCA4-specific shRNA (sh-PMCA4), GFP-PMCA4b and GFP-PMCA4b$^{LA}$-expressing MCF-7 cell lines. After 10 min incubation with WGA, cells were washed and transferred to a 37 °C incubator for 20 min. Dashed white squares indicate areas for magnification on (**d**). DAPI labels nuclei. **b** Yellow lines connect the center of the cell nucleus with WGA positive endocytic vesicles accumulated in the cytosol. **c** Intracellular distribution analysis of WGA positive vesicles. Red line indicates the distribution fit curve. Data were collected from 2 independent experiments, $n_{(MCF-7)} = 1214$, $n_{(sh-PMCA4)} = 922$, $n_{(PMCA4b)} = 1021$,

$n_{(PMCA4b^{LA})} = 1493$ vesicles from 15 to 28 cells, and analyzed as described in the Methods section. **d** Statistical analysis of the distribution of WGA positive vesicles is illustrated by a box plot diagram with medians, interquartile range boxes and whiskers. Interquartile rage boxes represent the middle 50% of the data. Whiskers represent the ranges for the bottom 25% and the top 25% of the data values. Data were the same as (**c**) and analyzed with Levene's test. $P$ values: **$P < 0.01$, ***$P < 0.001$; non-significant difference is not labeled. **e** Magnification of th**e** areas indicated by the dashed white squares on (**a**). Yellow arrows indicate colocalized WGA and PMCA4b positive vesicles.

MCF-7 cells. We have demonstrated that introducing PMCA4b to a BRAF mutant melanoma cell line resulted in a severe loss of integrin β4 expression[64]. These findings suggest that PMCA4 may play a role in lumen-formation by regulating integrin-mediated cell adhesion, highlighting the need for further investigation in this respect.

Silencing the single *D. melanogaster pmca* gene resulted in larval salivary gland lumen morphology degeneration (Fig. 9), suggesting an evolutionarily conserved role of PMCA proteins in lumen formation of glandular organs in vivo. It has been demonstrated that secretory activity requires cytoplasmic Ca$^{2+}$ oscillations[65–67], and local elevation of Ca$^{2+}$ concentration is necessary during fusion of secretory granules with the plasma membrane[68]. Our electron microscopy analysis of PMCA-deficient Drosophila salivary glands revealed a pronounced secretion defect and abnormal accumulation of secretory vesicles in vivo (Fig. 9g). Revealing the exact PMCA function in secretion requires, however, further investigation.

Recently, we demonstrated that PMCA4b over-expression induced cell polarization and actin cytoskeleton remodeling of BRAF mutant melanoma cells. Moreover, a profound change in cell culture morphology and F-actin rearrangement was detected in MCF-7 cells in response to changes in PMCA4b protein abundance[69]. Actin network formation and contraction are prerequisites of lumen formation[70] and several proteins involved in actin cytoskeleton dynamics are regulated by Ca$^{2+}$[71]. In vitro PMCA4 can directly

interact with G-actin and short actin oligomers. Both interactions activate PMCA ATPase function while interaction with F-actin inhibits its activity[72,73]. PMCA4b can also affect F-actin dynamics indirectly by reducing cytoplasmic Ca$^{2+}$ concentration that may prevent cortical F-actin degradation in response to an uncontrolled rise in Ca$^{2+}$[74]. It has been demonstrated that an enrichment of PMCA4 at the cell front was essential for the back to front Ca$^{2+}$ gradient in controlling F-actin dynamics at the lamellipodia during endothelial cell migration, however, the exact mechanisms for directing PMCA4 to the cell front has not been elucidated[75].

Our results highlight the importance of proper intracellular trafficking of PMCA4b in cell polarization and subsequent lumen formation and suggest that the small GTPase Arf6 may play a role in this process. We show that PMCA4b colocalizes both with the wild-type and a constitutively active form of Arf6, and that Arf6 inhibition interferes with PMCA4b recycling. Previous studies indicate that Arf6 plays a role in actin cytoskeleton remodeling and lumen formation in epithelial and endothelial cells[23,76–79] that is in good agreement with our current observation demonstrating that Arf6 inhibition prevents PMCA4b-induced pre-lumen formation in 2D cell cultures (Fig. 7e, f).

Our model shown in Fig. 10 suggests that elevated PMCA4b levels promote the polarization of luminal epithelial cells with proper plasma membrane localization of E-cadherin, Dlg1 and ezrin. PMCA4b is involved

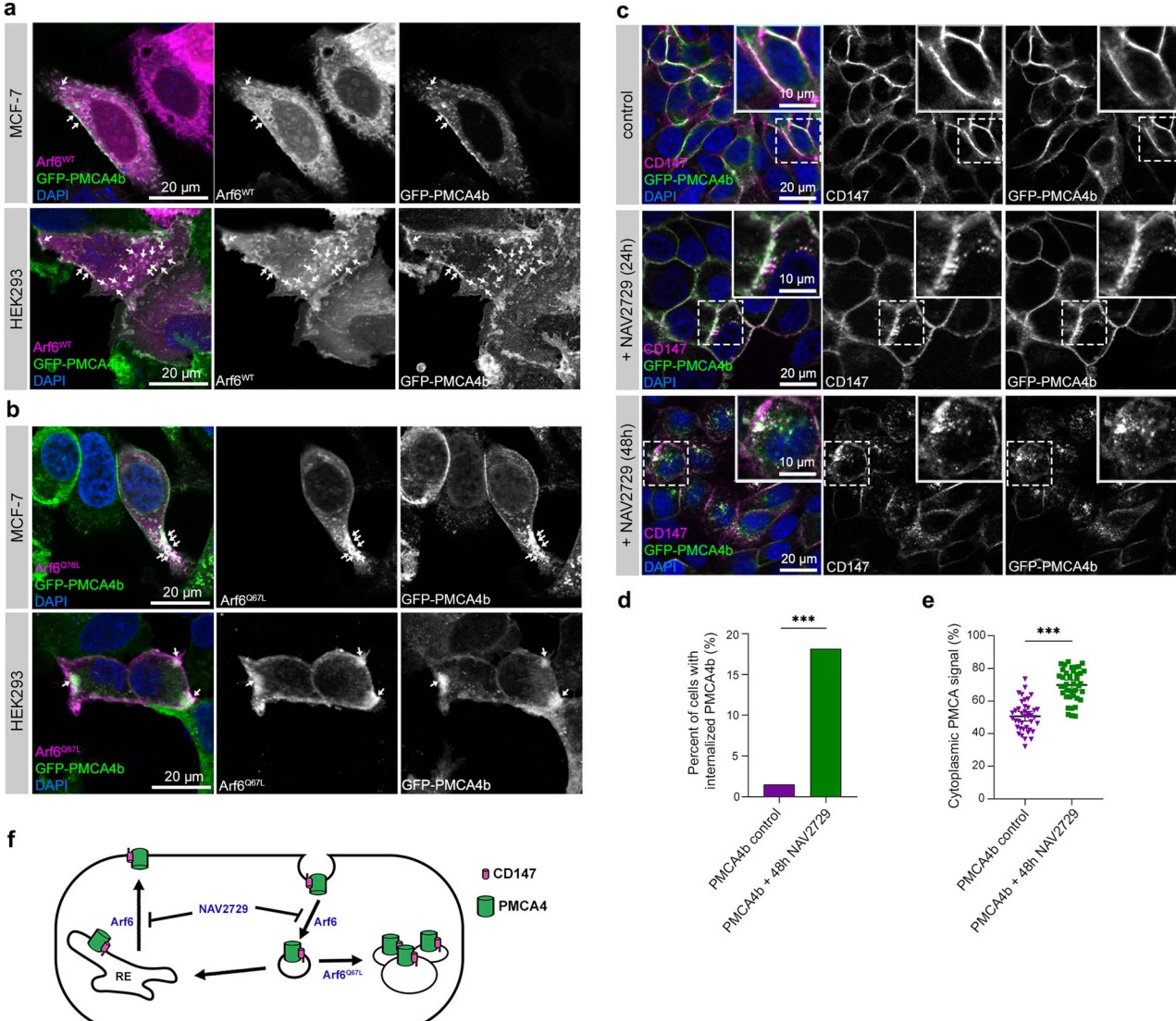

**Fig. 6 | Arf6 regulates endocytic recycling of PMCA4b. a, b** Transfection of GFP-PMCA4b-expressing MCF-7 and HEK-293 cell lines with wild-type (WT) and Q67L mutant form of Arf6. White arrows point to colocalizing areas. **c** CD147 immunostaining of non-treated and 24 or 48 h NAV2729-treated GFP-PMCA4b-expressing MCF-7 cells. Insets show magnified areas indicated by the dashed white rectangles. **d** Statistical analysis of the distribution of cells with internalized GFP-PMCA4b in control and after 48 h of NAV2729 treatment. Data were collected from 8 fields of view from 3 independent experiments; $n_{(PMCA4b\ control)} = 589$,

$n_{(PMCA4b\ +\ 48h\ NAV2729)} = 435$ cells. Data were analyzed with chi-square test, $p$ value: ***$p < 0.001$; non-significant difference is not labeled. **e** Statistical analysis of the GFP-PMCA4b signal distribution between cytoplasm and plasma membrane. Graph displays means with 95% confidence intervals; data were collected from 3 independent experiments, $n_{(PMCA4b\ control)} = 40$, $n_{(PMCA4b\ +\ 48h\ NAV2729)} = 40$ cells, and analyzed as described in the Methods section. Data were analyzed with unpaired $t$ test, $p$ value: ***$p < 0.001$. **f** A scheme depicting the role of Arf6 in PMCA4b trafficking. RE recycling endosome.

in dynamic endocytic vesicle trafficking towards the central pre-lumen area where it may participate in central actin stabilization at the early steps of lumen formation (Fig. 7b). This is followed by its recycling to the basolateral plasma membrane of epithelial cells in fully developed luminal structures (Fig. 8a). During these processes PMCA4b trafficking is regulated by Arf6, in which the recently described PDZ protein complex PLEKHA7–PDZD11 may play a role[80].

Loss of cell polarity and cell-cell contacts is a hallmark of epithelial malignancy[50,81]. Taken together with our observations on clinical outcome data (Fig. 1) this suggests that elevated levels of PMCA4b may be associated with a lower risk of cancer progression at early stages of luminal A and B1 breast cancer. It is important to note, however, that different cancer types have different molecular backgrounds and hence, PMCA4b may play distinct roles. For instance, in contrast to our findings in melanoma cells[11] and others in gastric cancer[62], PMCA4b knockdown reduced the migratory activity of pancreatic ductal adenocarcinoma MIA PaCa-2 cells and

increased their sensitivity to apoptosis[82]. It is worth mentioning that in pancreatic ductal cells PMCA4 localizes to the apical plasma membrane where it physically interacts with CFTR protein supporting its cell-type specific function[83]. We also show here that whereas PMCA4 abundance is associated with longer relapse-free patient survival in luminal LUMA (grade 2) and LUMB1 breast cancer subtypes, the opposite is observed for patients with HER2 positive LUMB2 breast cancer (Fig. 1d). This is not unexpected, since "tumor suppressors" like E-cadherin and Scrib may exhibit opposing functions across different tumor types, cancer stages, and/or at metastatic sites[81,84]. Likewise, PMCA4 may have cancer type-specific functions in various neoplastic processes, emphasizing the importance of precise histopathological and molecular characterization of individual tumor samples for optimal treatment choices.

In summary, our findings suggest that loss of PMCA4 is implicated in luminal type breast carcinomas, and that elevated PMCA4 levels are associated with longer relapse-free patient survival at the early onset of LUMA

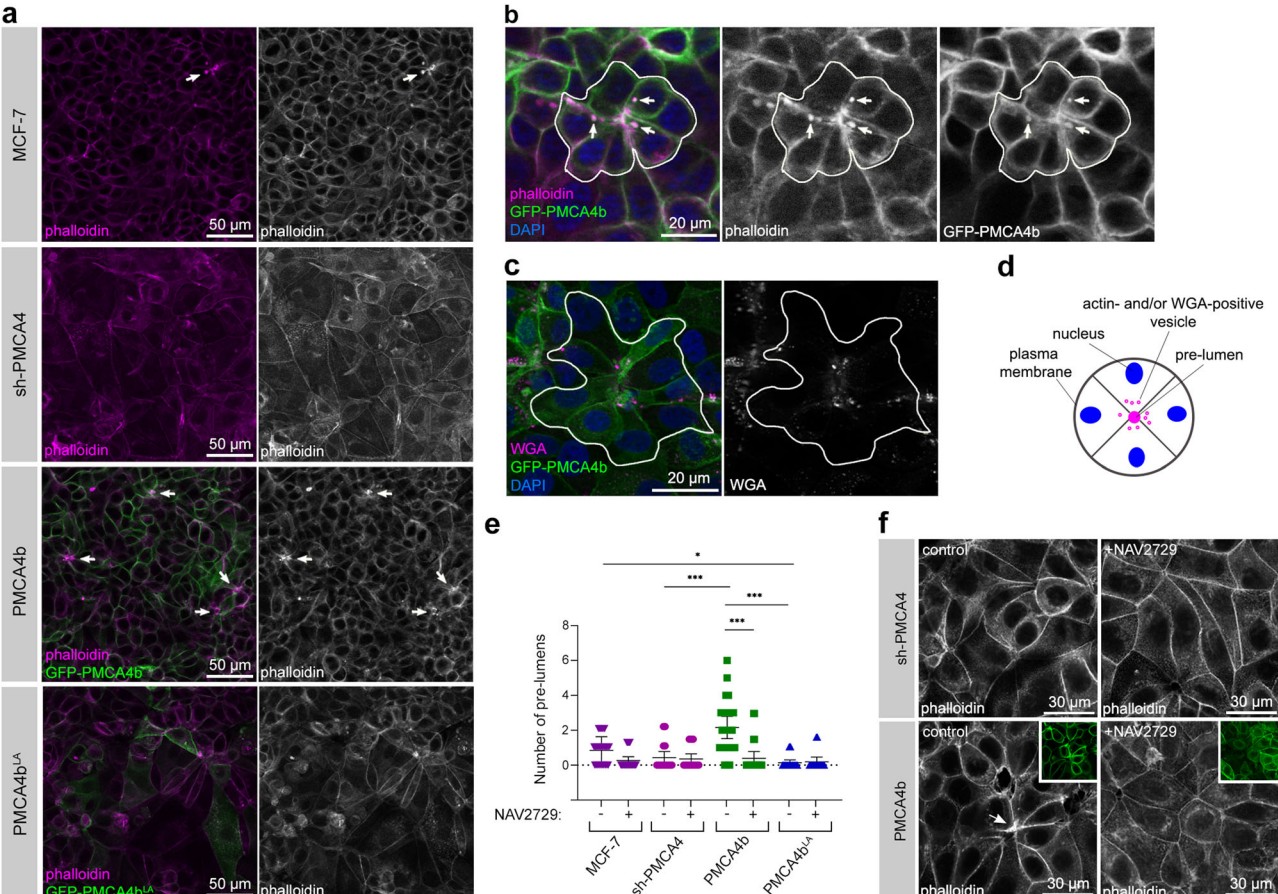

**Fig. 7 | PMCA4b promotes pre-lumen formation in 2D cultures of MCF-7 cells.**
**a** Phalloidin staining of parental, PMCA4 specific-shRNA (sh-PMCA4), GFP-PMCA4b and GFP-PMCA4b$^{LA}$-expressing MCF-7 cell lines. White arrows point to the pre-lumens. **b** Phalloidin staining of GFP-PMCA4b-expressing MCF-7 cells. The white line encircles a pre-lumen. White arrows point to phalloidin and PMCA4b double-positive vesicles. **c** WGA uptake assay on GFP-PMCA4b-expressing MCF-7 cells with 20 min incubation after WGA removal. Pre-lumen creating cells are encircled with white line. **d** A schematic figure of a typical pre-lumen structure in (**a**) 2D cell culture. **e** Statistical analysis of pre-lumen formation in control and NAV2729 treated parental, PMCA4 specific shRNA (sh-PMCA4), GFP-PMCA4b

and GFP-PMCA4b$^{LA}$-expressing MCF-7 cell lines. Graph displays means with 95% confidence intervals corrected with the total area of the images; data were collected from 2 independent experiments, $n_{(sh-PMCA4)} = 23$, $n_{(sh-PMCA4+NAV2729)} = 21$, $n_{(MCF-7)} = 26$, $n_{(MCF-7 + NAV2729)} = 22$, $n_{(PMCA4b)} = 25$, $n_{(PMCA4b+NAV2729)} = 23$, $n_{(PMCA4b^{LA})} = 23$, $n_{(PMCA4b^{LA} + NAV2729)} = 17$ images. Data were analyzed with Kruskal–Wallis and Dunn's multiple comparisons tests, adjusted $p$ values: $^{ns}p \geq 0.05$, $^*p < 0.05$, $^{**}p < 0.01$ $^{***}p < 0.001$; non-significant differences are not labeled. **f** Phalloidin staining of control and NAV2729 treated PMCA4 specific shRNA (sh-PMCA4) and GFP-PMCA4b-expressing MCF-7 cell lines. Scaled-down insets show the GFP-PMCA4b signal. The white arrowhead points to a pre-lumen.

---

and LUMB1 breast cancer cases. Our results may indicate that PMCA4b is a novel polarity protein and point towards its potential evolutionarily conserved function in the formation of luminal structures in exocrine glandular organs.

## Materials and Methods
### Tissue samples
One hundred nine formalin fixed paraffin embedded HR$^+$ (LUMA $n = 39$; LUMB1 $n = 50$; LUMB2 $n = 20$ study cohort) and 4 normal breast tissue samples obtained after breast reduction surgery were selected (Supplementary Data). The cases were diagnosed at the Department of Pathology, Forensic and Insurance Medicine, Semmelweis University, Hungary between 2000 and 2010. Clinicopathological data of the patients were obtained from the files of the Semmelweis University, 2nd Dept. of Pathology and from the Semmelweis University Health Care Database with the permission of the Hungarian Medical Research Council (ETT-TUKEB 14383/2017). All ethical regulations relevant to human research participants were followed. Due to the retrospective nature of the study, the need of informed consent was waived by the Hungarian Medical Research Council (ETT-TUKEB 14383/2017). Surrogate breast carcinoma subtype was defined based on four (estrogen receptor (ER), progesterone receptor (PgR), Ki-67 index (marker of proliferation) and human epidermal growth factor

receptor-2 (HER2) immunohistochemical markers and according to the 2013 St. Gallen Consensus Conference recommendations[85].

Luminal A (LUMA) tumors are defined as ER and PgR positive, HER2 negative, Ki-67 "low" (Ki-67 < 20%) tumors, Luminal B-HER2 negative (LUMB1) tumors as ER positive, HER2 negative and Ki-67 "high" (≥20%) and/or PgR "negative or low" (PgR cut-point = 20%) and Luminal B-HER2 positive (LUMB2) as ER positive and HER2 overexpressed or amplified.

### Immunohistochemistry to evaluate PMCA4 expression in breast tissue samples
Immunohistochemistry (IHC) staining on all human tissue samples was performed and evaluated at the Department of Pathology, Forensic and Insurance Medicine, Semmelweis University. Cut-off values for estrogen receptor (ER) and progesterone receptor (PgR) positivity were 1% of tumor cells with nuclear staining and defined as positive or negative. Human erb-b2 receptor tyrosine kinase 2 (HER2) status was determined either as protein overexpression or HER2 gene amplification detected by fluorescence in situ hybridization (FISH). Data were analyzed by Kruskal–Wallis and Mann–Whitney tests for comparison of quantitative variables by the statistical software SPSS V.25 (IBM Corp.).

PMCA4 IHC was performed with an automated immunostainer system (Ventana Benchmark Ultra, Roche Diagnostics) according to the

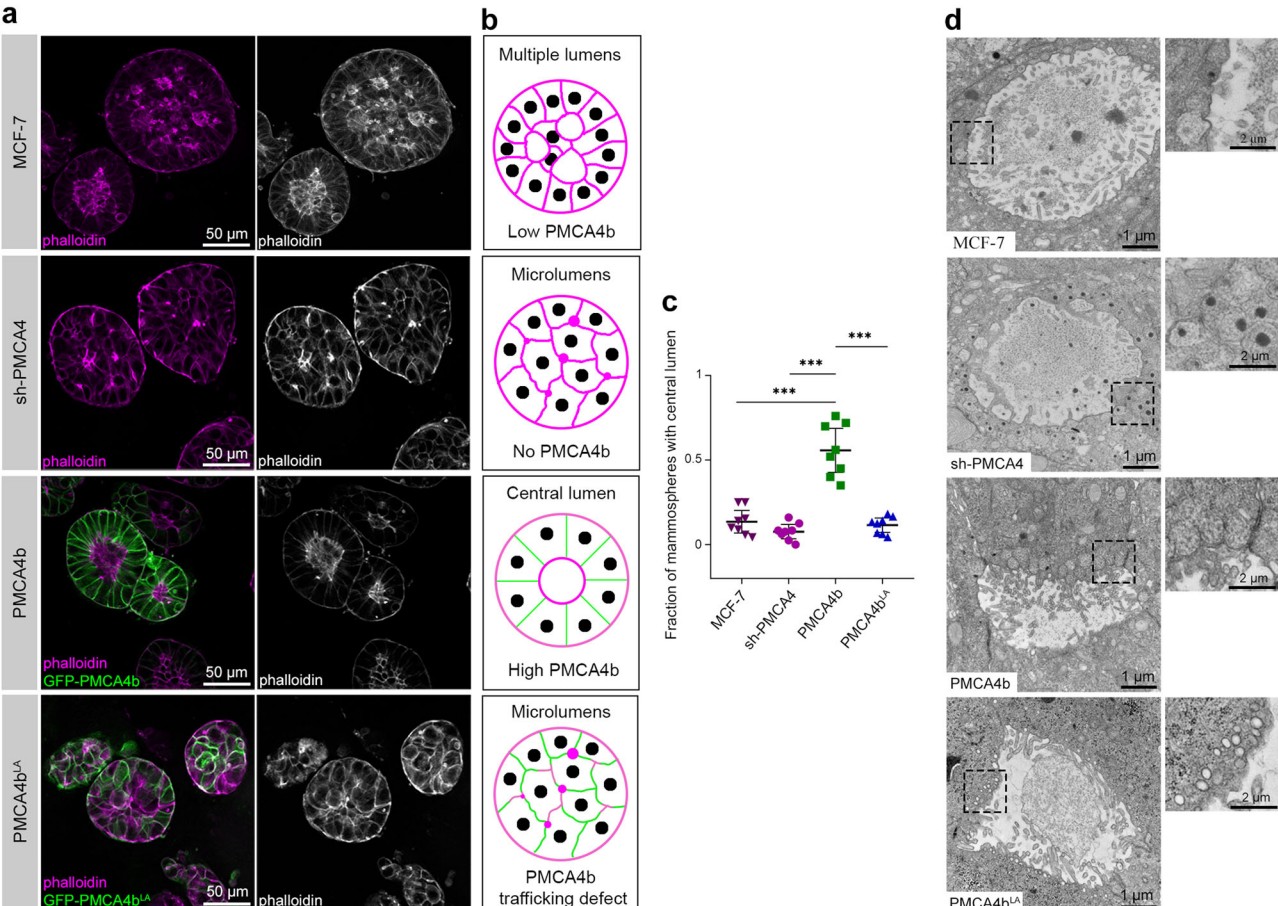

**Fig. 8 | PMCA4b promotes central lumen formation in 3D cultures of MCF-7 cells. a** Phalloidin staining of mammospheres formed by parental, PMCA4-specific shRNA (sh-PMCA4), GFP-PMCA4b and GFP-PMCA4b$^{LA}$-expressing MCF-7 cell lines cultured in Matrigel for 10 days. **b** Schematic figures demonstrating structures of mammospheres based on results on (**a**). **c** Statistical comparisons of the ratios of mammospheres with central lumen in 3D cultures between parental, PMCA4-specific shRNA (sh-PMCA4), GFP-PMCA4b- and GFP-PMCA4b$^{LA}$-expressing MCF-7 cell lines cultured in Matrigel. Graph displays means with 95%

confidence intervals, $n = 8$ images with 10–15 mammospheres per image for each cell type. Data were analyzed with ordinary one-way ANOVA and Tukey's multiple comparisons tests, adjusted $p$ values: $^{ns}p \geq 0.05$, $*p < 0.05$, $**p < 0.01$, $***p < 0.001$, non-significant differences are not labeled. **d** Transmission electron microscope images of 14 days old mammospheres formed by parental, PMCA4-specific shRNA (sh-PMCA4), GFP-PMCA4b and GFP-PMCA4b$^{LA}$-expressing MCF-7 cell lines cultured in Matrigel. Insets show magnified areas indicated by the dashed black squares.

---

manufacturer's instructions. Antigen retrieval was performed with ULTRA CC1 antigen retrieval solution for 90 minutes at 95 °C. The primary anti-PMCA4 JA9 mouse monoclonal antibody (P1494, Merck) was used at a 1:200 dilution at 42 °C for 32 min. For antibody visualization, the UltraView DAB Detection kit (Ventana) was applied. PMCA4 stained slides were scanned with slide-scanner (PANNORAMIC® 1000 DX, 3DHISTECH Ltd., Hungary). PMCA4 IHC reaction was semi-quantitatively evaluated on digitized slides: (1) IHC score was defined as the percentage of positive epithelial cells counted on average in 3–5 high-power fields, (2) membrane positivity with or without cytoplasmic reaction was assessed. PMCA4 IHC scoring was performed on digitized slides by using the slide-viewing software CaseViewer (CaseViewer 2.3.0.99276, 3DHISTECH Ltd., Hungary).

### Reagents and treatments
Phalloidin-TRITC (P1951, Sigma-Aldrich) and NAV2729 (SML2238, Sigma-Aldrich) were dissolved in dimethyl sulfoxide and stored at −20 °C at a concentration of 80 µM and a 4.5 mM, respectively. Phalloidin-TRITC was used for F-actin labeling on confluent 2D cultures and 3D mammospheres of MCF-7 cells at a concentration of 80 nM. In the experiments studying the role of Arf6, MCF-7 cells were treated with 6.75 µM NAV2729 for 24 or 48 h. To label nuclei 2-(4-amidinophenyl)-1H-indole-6-carboxamidine (DAPI) was used at a concentration of 0.5 µM. For *D. melanogaster*

salivary gland staining Rhodamine Phalloidin (ab235138, Abcam) was used at a 1000× dilution in PBS according to the manufacturer's instructions.

### Cell culture
MCF-7 (cat. number: HTB-22) and HEK-293T (cat. number: CRL-1573) cell lines were purchased from the American Type Culture Collection, T47D cells were purchased from NCI Development Therapeutics Program (DCTD Tumor Repository, National Cancer Institute at Frederick, MD). MCF-7 and HEK-293T cells were cultured in Dulbecco's modified Eagle's medium (DMEM) (DMEM-HXA, Capricorn Scientific), while T47D cells in RPMI 1640 Medium (RPMI-A, Capricorn Scientific) supplemented with 10% heat-inactivated fetal bovine serum (FBS; ECS0180L, Euroclone), 2mM L-glutamine (STA-B, Capricorn Scientific), 100 µ/ml Penicillin and 100 µg/ml streptomycin (PS-B, Capricorn Scientific) in a humidified 5% $CO_2$ incubator at 37 °C. In order to maintain the stable expression of PMCA4b transgenes and of the shRNA construct, 100 ng/ml puromycin dihydrochloride (sc-108071, Santa Cruz Biotechnology) was added to the culture media. Cell cultures were routinely tested for mycoplasma contamination.

### Generation of stable cell lines
PMCA4-specific shRNA expressing MCF-7 cells were generated by transfecting with PMCA4-specific shRNA plasmid (sc-42602-SH, Santa Cruz Biotechnology, Santa Cruz, CA, USA) that contains puromycin resistance

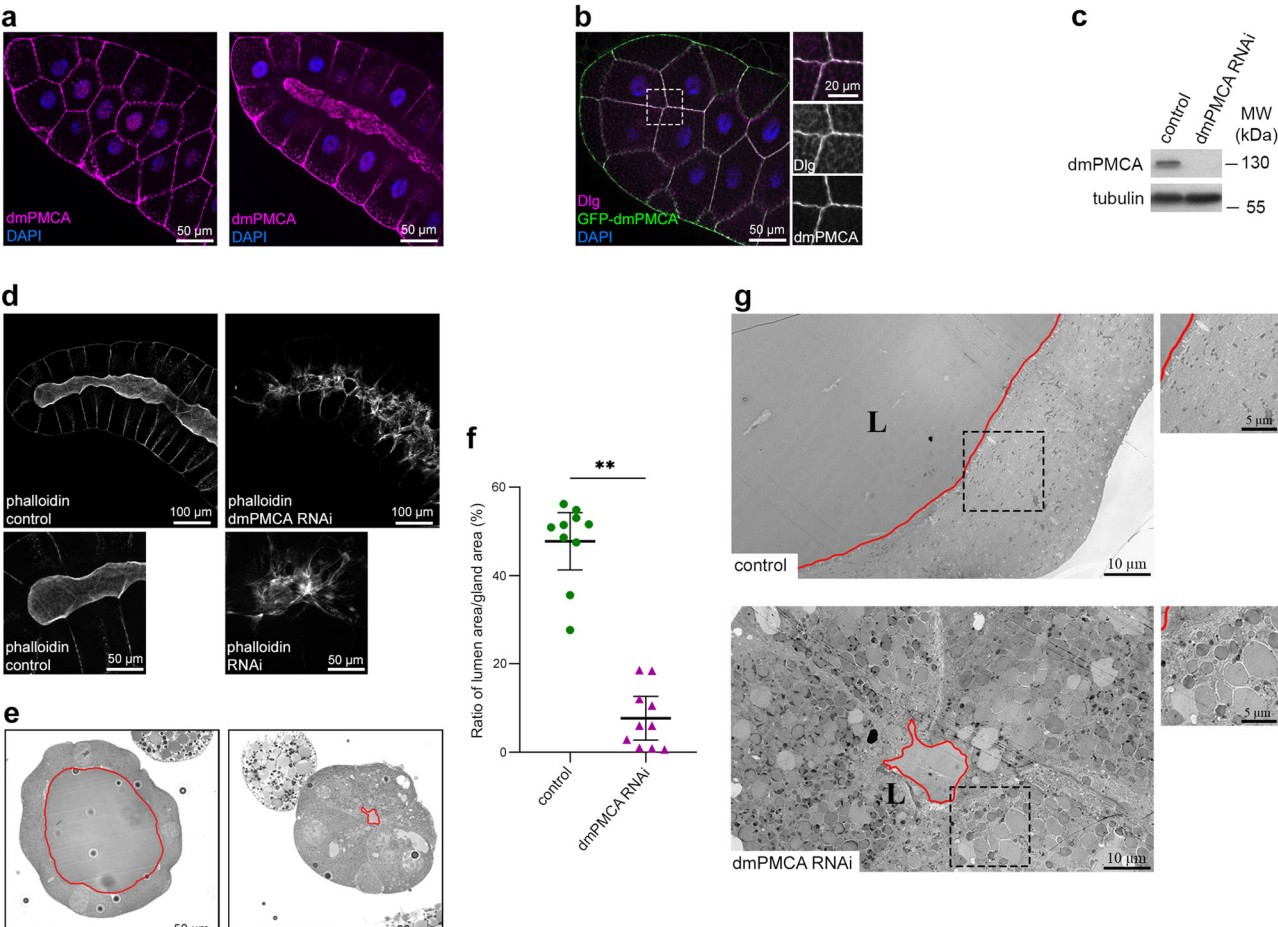

**Fig. 9 | *Drosophila melanogaster* (dm)PMCA regulates lumen size and morphology in larval salivary gland. a** DmPMCA immunostaining of the salivary gland of a 3rd stage larva. On the left: upper cell layer section; on the right: middle section. **b** Co-immunostaining of *D. melanogaster* Dlg and (dm)PMCA of the salivary gland of a 3rd stage larva. **c** Western blot analysis of dmPMCA expression in control and ubiquitous *daughterless* promoter-driven dmPMCA-specific RNAi-expressing 3rd stage *D. melanogaster* larvae. **d** Phalloidin staining of control and salivary gland-specific *forkhead* promoter-driven dmPMCA RNAi-expressing 3rd stage larvae. Images in the lower row show distal parts of the lumen in the same salivary gland as above using higher magnification. **e** Toluidine blue-stained semi-thin cross-sections of salivary glands from 3rd stage control and *forkhead* promoter-driven dmPMCA specific RNAi-expressing larvae. The lumen is outlined by red line. **f** Statistical comparisons of the lumen area ratios on semi-thin cross-sections. Graph displays means with 95% confidence intervals, *n* = 10 animals. Data were analyzed with Mann–Whitney test; *p* values: $^{ns}p \geq 0.05$, $**p < 0.01$. **g** Transmission electron microscope images from cross-sections of salivary gland of control and salivary gland specific *forkhead* promoter-driven dmPMCA RNAi-expressing 3rd stage larvae. Magnified areas indicated by the dashed black rectangles are shown on the right. The lumen is outlined by red line. L lumen.

gene using FuGENE HD transfection reagent (Promega, Madison, WI, USA) according to the manufacturer's instructions. The culture medium was changed to a selection medium containing puromycin dihydrochloride (1 μg/ml) after 48 h. Cells were grown in puromycin supplemented medium for 2 weeks and PMCA4b silencing was confirmed by western blot[69]. GFP-PMCA4b and GFP-PMCA4b[LA]-expressing MCF-7, GFP-PMCA4b-expressing HEK-293T and GFP-PMCA4b-expressing T47D cell lines were generated by stable transfection of the cells with *SB-CAG-GFP-PMCA4b-CAG-puromycin* or *SB-CAG-GFP-PMCA4b-LA-CAG-puromycin* constructs that contain a Sleeping Beauty transposon system. Cells were transfected with the transposon constructs and Sleeping Beauty 100× (SB100×) transposase plasmid in a 1:10 ratio using FuGENE HD transfection reagent. Forty eight hours after transfection, the medium was changed to a puromycin dihydrochloride containing (1 μg/ml) selection medium. Selection was continued until all non-transfected cells died[11,69]. Created cell lines are not authenticated.

### Transgenic *Drosophila melanogaster* lines
Flies were raised at 25 °C on standard cornmeal, yeast and agar-containing medium. The *w[1118]* strain (FlyBase ID: FBst0003605) was used as control and *da-Gal4* fly line (exact genotype: *w\*;P(UAS-da.G)52.2*, FlyBase ID: FBst0051669) were obtained from the Bloomington Drosophila Stock Center (BDSC), the *PMCA[2165R-3] RNAi* line (stock ID: 2165R-3) was obtained from the Fly Stocks of National Institute of Genetics (Nig-Fly) and the PMCA-GFP line was obtained from Kyoto Drosophila Stock Center (exact genotype: *w1118; PBac(681.P.FSVS-1)PMCACPTI001995*, DGRC number: 115256). The *fkh-Gal4* fly line was provided by Eric H. Baehrecke[86] (University of Massachusetts Medical School, Worcester, MA). Exact phenotypes of Drosophila larvae, which were used for experiments in Fig. 7, were indicated in Supplementary Table.

### Ca$^{2+}$ signal measurement
MCF-7 cells were seeded in 8-well chamber (155411, Nunc) and 48 h later were transfected with CMV-R-GECO1 (a gift from Robert Campbell[87] (32444, Addgene) using FuGENE HD transfection reagent (Promega Corporation), according to the manufacturer's recommendations. Forty eight hours after transfection, DMEM was replaced by Hanks' Buffered Salt Solution (HBSS, Thermo Fischer 88284) supplemented with 10 mM HEPES (pH 7.4) and 2 mM CaCl$_2$. After 2 minutes baseline imaging, 10 μM ATP was added to the cells. Images were acquired every 0,4 s for 10 min. The

https://doi.org/10.1038/s42003-025-07814-5                                                              **Article**

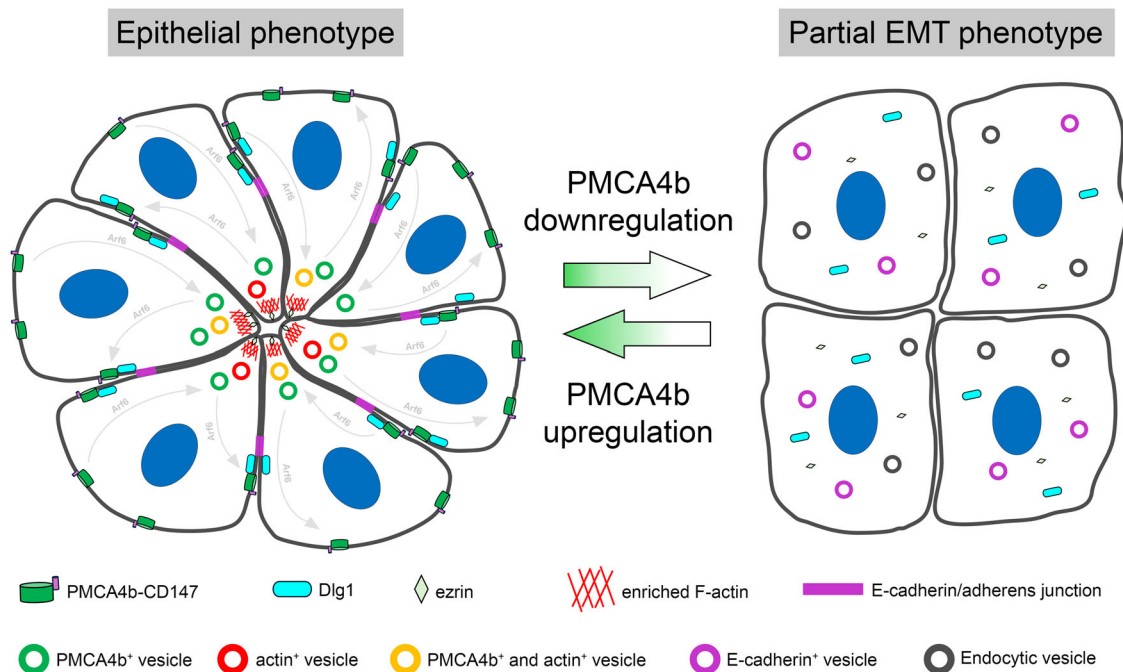

**Fig. 10 | A model on the role of PMCA4b in luminal epithelial cell polarization and lumen formation.** PMCA4b facilitates the lumen forming capability of luminal epithelial cells through contributing to the polarized distribution of ezrin, and the plasma membrane localization of E-cadherin and Dlg1. During (pre-)lumen formation dynamic vesicular trafficking toward the emerging lumen area includes trafficking of PMCA4b through its Arf6-mediated recycling to the basolateral plasma membrane compartment. Losing PMCA4b, epithelial cells lose their polarized phenotype and acquire partial EMT characterized by cytoplasmic localization of E-cadherin, Dlg1 and ezrin.

relative fluorescence was calculated as $F/F_0$ (where $F_0$ was the average of 30 s baseline fluorescence). Time-lapse sequences were recorded with a Nikon Eclipse Ti2 confocal microscope with a Plan-Apochromat 20×/0.75 Air objective with iMSPECTOR software (Abberior Instruments GmbH) at room temperature.

**Immunofluorescence**
**Immunocytochemistry of human cells.** Cells were seeded in removable 8-well chamber (80841, Ibidi) and 96 h later rinsed with phosphate buffered saline (PBS) at 37 °C, and fixed with 4% PFA diluted in PBS for 10 min. After rinsing twice and washing twice with PBS for 5 min, cells were permeabilized for 10 min. with 0.2% Triton X-100 dissolved in PBS. After rinsing twice and washing twice with PBS, the cells were treated with Image-iT™ FX Signal Enhancer (I36933, Invitrogen) for 30 min to reduce non-specific binding of fluorophores. After rinsing twice with PBS, the samples were incubated in blocking solution containing 2 mg/ml bovine serum albumin (A7030, Sigma-Aldrich), 1% gelatin from cold water fish skin (G7765, Sigma-Aldrich), 0.1% Triton X-100 (X100PC, Sigma-Aldrich) and 5% goat serum (G9023, Sigma-Aldrich) diluted in PBS, for 1 h at room temperature (RT). Samples were then incubated in the following primary antibodies diluted in blocking solution: anti-E-cadherin (24E10, CST 3195, Cell Signaling, dilution 1:200), anti-ezrin (E1281, Sigma-Aldrich, dilution 1:200), anti-Dlg1/SAP97 (sc-25661, Santa Cruz Biotechnology, dilution 1:100), anti-CD147/basigin (10186-R125, Sino Biological, dilution 1:200), or anti-HA (H3663, Sigma-Aldrich, dilution 1:500). After rinsing twice and washing twice for 5 min with PBS, the Alexa Fluor 594-conjugated goat anti-rabbit and anti-mouse IgG secondary antibodies (A-11012 and A-11005, Invitrogen, dilution 1:500) were added. Samples were rinsed twice and washed twice with PBS for 5 min and the nucleus was labeled with DAPI (details in section *Reagents and treatments*). The silicon part of the chamber was then removed, and the samples were mounted with Ibidi Mounting Medium (50001, Ibidi).

**Phalloidin staining of human cells.** MCF-7 and T47D cells were seeded, grown, fixed, and permeabilized as described in the above section. 80 nM Phalloidin-TRITC (P1951, Sigma-Aldrich) was added to the cells after permeabilization and incubated for 1 h. Then the cells were washed with PBS, DAPI labeled and mounted as described above.

**IHC staining of Drosophila larval salivary gland.** Salivary glands were dissected from late L3 stage Drosophila larvae in PBS and pre-permeabilized for 20 s in 0.05% Triton X-100 diluted in PBS. Salivary glands were then fixed for 40 min at room temperature in 4% PFA diluted in PBS, rinsed twice and washed twice with 0.1% Triton X-100 diluted in PBS (PBST) for 15 min and incubated for 40 min at room temperature (RT) in blocking solution that contained 10% FBS and 0.1% Triton X-100 diluted in PBS; then salivary glands were incubated for 24 h at 4 °C with the following primary antibodies: anti-*pan*-PMCA (5F10, MABN1802, Sigma-Aldrich, dilution 1:250), anti-Dlg (4F3, Developmental Studies Hybridoma Bank, dilution 1:100), anti-GFP (A10262, Invitrogen, dilution 1:1000) diluted in blocking solution. After rinsing twice and washing twice the samples with PBST for 15 min, they were incubated with the secondary antibodies diluted in blocking solution: Alexa Fluor 488-conjugated goat anti-chicken IgY (A-11039, Invitrogen, dilution 1:1000) and Alexa Fluor 594-conjugated goat anti-mouse IgG (A-11020, Invitrogen, dilution 1:1000). After rinsing twice and washing twice the samples with PBST for 15 min, nuclei were labeled with DAPI (as in section *Reagents and treatments*) and samples were mounted with Vectashield (H-1000-10, Vector Laboratories).

**Phalloidin staining of Drosophila larval salivary glands.** Salivary glands were dissected, permeabilized and fixed the same way as described in the previous section. For F-actin labeling, the samples were incubated with Rhodamine Phalloidin (details in section *Reagents and treatments*) for 1 h, then washed, nuclei labeled with DAPI and mounted the same way as described for IHC.

## Imaging

Fluorescent images were obtained with an Axio Imager M2 microscope (ZEISS) with an ApoTome2 grid confocal unit (ZEISS) using a Plan-Apochromat 63×/1.4 Oil (ZEISS) objective for human cells, a Plan-Apochromat 40×/0.95 Air (ZEISS) objective for semi-thin sections of Drosophila salivary glands, a Plan-Neofluar 20×/0.5 Air objective for mammospheres and Drosophila salivary glands with an Orca Flash 4.0 LT sCMOS camera (Hamamatsu Photonics) using the Efficient Navigation 2 software (ZEN 2) (ZEISS). Images in Fig. 6a were obtained with LSM 710, AxioObserver confocal laser-scanning microscope (ZEISS) using a Plan-Apochromat 63×/1.4 Oil DIC M27 objective (ZEISS). Images in Fig. 7f and Supplementary Fig. 5 were obtained with a Nikon Eclipse Ti2 confocal microscope with a Plan-Apochromat 60×/1.4 Oil objective with iMSPEC-TOR software (Abberior Instruments GmbH).

Time-lapse video imaging was performed on an inverted Nikon Eclipse Ti2 microscope with 488 nm and 638 nm laser lines for the Yokogawa CSU-W1 spinning disc scan head equipped with two back illuminated Photo-metrics Prime BSI scientific CMOS cameras for detection. Green and red fluorescence were recorded using 525/50 nm and 641/75 nm emission filters. For the recording, we used the NIS Elements AR 5.4 software with a dimension of 1024 × 1024 pixels and bit depth of 16 bits. For acquisition, we used a 40× ApoLambda/NA1.15 water-dipping objective. The videos were captured for 25 min. Final movies were created by the ImageJ 1.54 f software[88].

## Growing and staining mammospheres

Five thousand cells from the parental, GFP-PMCA4b, GFP-PMCA4b[LA] and PMCA4-specific shRNA-expressing MCF-7 lines were mixed with cold Matrigel (Corning Matrigel, DLW356231, Sigma-Aldrich) and 15 µl of cell-Matrigel mixtures were added to removable 8-well chambers (80841, Ibidi) and incubated at 37 °C for Matrigel polymerization. After 15 min, FBS-supplemented DMEM was added to the wells, and cells were grown for 10 days until they formed mature mammospheres. For F-actin staining the samples were washed with PBS at 37 °C, fixed for 40 min with 4% PFA dissolved in PBS, permeabilized for 20 min with 0,2% Triton X-100 in PBS, and after washing TRITC-Phalloidin was added for F-actin staining (see further details in section *Reagents and treatments*). Samples were mounted with Ibidi Mounting Medium.

## Transfection

GFP-PMCA4b-expressing MCF-7 ($5 \times 10^4$) and HEK-293T ($7 \times 10^4$) cells were seeded in removable 8-well chambers (80841, Ibidi). Next day cells were transiently transfected with the following plasmids: *pcDNA3/hArf6(WT)-mCherry* (Addgene plasmid #79422) and *pcDNA3/hArf6(Q67L)-HA* (Addgene plasmid # 79425; the plasmids were a gift from Kazuhisa Nakayama[89]) using the FuGENE HD Transfection Reagent (E2311, Promega) according to the manufacturer's instructions. Twenty four hours after transfection, cells were fixed, permeabilized, stained with DAPI and mounted. In the case of Arf6[Q67L] the cells were fixed and labeled with anti-HA antibody on the next day (details are the same as described in the section *Immunofluorescence*).

## WGA uptake assay

$5 \times 10^4$ cells from the parental, GFP-PMCA4b, GFP-PMCA4b[LA] and PMCA4-specific shRNA-expressing MCF-7 lines were seeded in removable 8-well chambers (80841, Ibidi) and were grown for 5 days in FBS-supplemented DMEM. Cells were rinsed with cold (4 °C) HBSS (Hanks' Balanced Salt solution; H15-009, PAA Laboratories) twice and incubated with WGA (W21404, Life Technologies) at a concentration of 5 µg/ml in HBSS for 10 min at room temperature. Unbound WGA was removed by rinsing the cells twice with PBS. The cells were then fixed, permeabilized and stained with DAPI, as described in the section *Immunofluorescence*.

## Sulforhodamine B (SRB) Assay

MCF-7 cells at densities indicated in Supplementary Fig. 2c were seeded in a 96-well plate and cultured for 96 h. Then cells were fixed with 10%

trichloroacetic acid at 4 °C for 1 h and washed with distilled water. After air dried overnight, 0.4% SRB was added to the wells and incubated for 15 min at room temperature. Excess stain was removed by washing with 1% acetic acid solution, and the bound SRB dye was solubilized in 10 mM Tris base solution for 10 min at RT with agitation. Optical density was measured at 570 nm using a microplate reader (EL800, BioTek Instruments, Winooski, VT, USA).

## Western blot analysis

The total proteins of MCF-7, A375 and T47D cells were extracted by the addition of 6% trichloroacetic acid[57]. *D. melanogaster* third-instar larvae were collected 1:1 Laemmli-PBS buffer and heated for 5 min at 100 °C, homogenized in Laemmli buffer, and centrifuged to separate the protein-containing supernatant. Equal amounts of proteins were loaded on 10% polyacrylamide gel, electrophoresed and electroblotted onto polyvinylidene difluoride (PVDF) membrane. Primary antibodies used for immunostaining were: anti-PMCA4 (JA9, P1494, Sigma-Aldrich, dilution 1:1000), anti-*pan*-PMCA (5F10, MABN1802, Sigma-Aldrich, dilution 1:2500), anti-E-cadherin (CST 3195, Cell Signaling, dilution 1:1000), anti-N-cadherin (sc-8424, Santa Cruz Biotechnology, dilution 1:500), anti-vimentin (M7020, Dako Products, dilution 1:1000), anti-β-actin (A1978, Sigma-Aldrich, dilution 1:2000), anti-Dlg1/SAP97 (sc-25661, Santa Cruz Biotechnology, dilution 1:1000), anti-CD147/basigin (10186-R125, Sino Biological, dilution 1:1000), anti-β-tubulin (ab6046, Abcam, dilution 1:1000). Horseradish peroxidase-conjugated secondary antibodies were used (715-035-151 and 711-035-152, Jackson ImmunoResearch, dilution 1:10,000) for detection with Pierce ECL Western Blotting Substrate (Thermo Scientific) and luminography on CL-XPosure Film (Thermo Scientific) or Amersham Hyperfilm ECL (GE Healthcare) films.

## Transmission electron microscopy (TEM)

MCF-7 cells were grown in Matrigel as described above for 14 days to create mammospheres. After the medium was removed, mammospheres were washed with FBS-free DMEM and fixed in a 3.2% methanol-free, ultrapure formaldehyde, 0.25% glutaraldehyde, 0.029 M sucrose, 0.002 M $CaCl_2$ and 0.1 M sodium cacodylate-containing fixing solution for 24 h at 4 °C. Samples were then washed for $2 \times 5$ min with 0.1 M sodium cacodylate, post-fixed in 1% ferrocyanide-reduced $OsO_4$ diluted in 0.1 M sodium cacodylate, pH 7.4, at 4 °C for 1 hr and washed again for $2 \times 5$ min with 0.1 M sodium cacodylate. For contrasting, samples were washed for $2 \times 5$ min with ultrapure distilled water and incubated in 1% uranyl-acetate diluted in distilled water for 30 min. Then samples were dehydrated by using a graded series of ethanol and were embedded in Spurr low viscosity epoxy resin (EM0300, Sigma Aldrich). Seventy nanometer ultrathin sections were made with an Om U3 microtome (Reichert Austria), collected on copper grids (AGG214, Agar Scientific) and stained in Reynold's lead citrate. For examination we used a transmission electron microscope (JEM-1011; JEOL, Tokyo, Japan) equipped with a digital camera (Morada; Olympus) using iTEM software (Olympus).

## Production of semi-thin cross-sections of *Drosophila melanogaster* larvae

Salivary glands were dissected from late L3 stage Drosophila larvae then fixed in a 3.2% methanol free, ultrapure formaldehyde, 0.25% glutaraldehyde, 0.029 M sucrose, 0.002 M $CaCl_2$ and 0.1 M sodium cacodylate containing fixing solution for 24 h at 4 °C. Samples were then immobilized in 1.5% agar dissolved in distilled water (this helps to keep orientation and integrity of the samples), and embedded in Spurr resin the same way as described in section *Transmission electron microscopy*. 0.5-1 µm semi-thin sections were made with an Om U3 microtome (Reichert Austria) and stained with a 1% toluidine blue, 1% Azure II, 1% borax and 25% sucrose containing solution.

## Image analysis, statistics and reproducibility

IHC data were analyzed by Kruskal–Wallis and Mann–Whitney tests for comparison of quantitative variables by statistical software SPSS V.25 (IBM

Corp.).To compare the prognostic impact of *ATP2B4* (PMCA4) and *ATP2B1* (PMCA1) expression in different breast carcinoma subtypes the publicly available database KM Plotter online tool (http://www.kmplot.com) was used. Only JetSet best probe set, and best cutoff values were selected, and Relapse Free Survival (RFS) plots were visualized by the Kaplan–Meier plotter using microarray data analysis[90]. Data for PMCA4 (NP_001675.3:S328), PMCA1 (NP_001673.2:S1182) protein expression in normal and luminal breast cancer subtypes were derived from the CPTAC and analyzed by using the UALCAN web portal[29] (https://ualcan.path.uab.edu/analysis-prot.html). Student's *t* test *p*-values were considered significant below 0.05. Data for mRNA level comparison of *ATP2B1*, *ATP2B2*, and *ATP2B*4 were derived from TCGA of the CPTAC database and analyzed by Student's *t* test.

To evaluate the number of E-cadherin-positive vesicles, the fluorescent images from original, unmodified single focal planes were analyzed with the ImageJ 1.54 f software. The fluorescence signal was set by auto threshold, transformed into binary format and particles were counted with the Analyze Particles command, in which particle circularity was set between 0.5 and 1 to exclude the membranous signal. Data were normalized to the total cell number determined by the nuclear dye DAPI image. The evaluation of Dlg1-positive vesicles followed the same procedure as for E-cadherin except that background subtraction was performed using the *Substract background* command in ImageJ 1.54 f, with a rolling ball radius set to 4.0. Intensity profiles for E-cadherin and Dlg1 image analysis were created by using the ZEN 2 Software for obtaining intensity data and GraphPad Prism 8.0 (GraphPad Software, Boston, Massachusetts USA, www.graphpad.com) for creating the graphs. Intensity was measured along a line that crosses the plasma membrane of two cells and avoids nuclei. Outlines, representing E-cadherin and Dlg1 positive cell compartments were created using ImageJ 1.54 f by setting the auto threshold command and using *Analyze Particles* function to display *bare outlines*. Number of pre-lumens in 2D cell cultures and lumens in 3D cultures were quantified manually. To quantify the size of the lumens in Drosophila salivary glands, we measured the total area of the gland, as well as the lumen area in the cross-section images using ZEN 2 software, and their ratio was determined by dividing the lumen area with the total area of the gland. For $Ca^{2+}$ signaling analysis the area under curve and first peak maximum values were calculated in GraphPad Prism 8.0. For analyzing cytoplasmic PMCA4b localization after NAV2729 treatment total and the cytoplasmic GFP-PMCA4b fluorescence signal intensities of individual cells were determined using the ImageJ software 1.54 f (Supplementary Fig. 3). The ratio of cytoplasmic signal was calculated by dividing the cytoplasmic signal with the total signal.

Data derived from the above fluorescent imaging analyses were imported into the GraphPad Prism 8.0 software and tested for normality of data distribution using D'Agostino & Pearson and Shapiro-Wilk tests. When data showed normal distribution with both tests, we used unpaired *t*-test or one-way ANOVA and Tukey's multiple comparisons tests; when data showed non-normal distribution, we used Mann–Whitney or Kruskal–Wallis and Dunn's multiple comparisons tests.

To define the percentage of polarized MCF-7 cells in a 2D cell cultures, the number of cells with either asymmetric or symmetric distribution of the apical marker ezrin was quantified manually. The cells were sorted based on their ezrin distribution into polarized and non-polarized groups. To analyze the percentage of cells with internalized PMCA4b in response to NAV2729 treatment, cells were sorted into groups based on their PMCA4b localization (plasma membrane or vesicular) manually. From both analyses the number of cells was collected into contingency tables and data were analyzed by chi-square test.

To analyze planar distribution of WGA positive vesicles in MCF-7 cells we defined the center of nucleus with the Centroid command of the ImageJ software 1.54 f. The center of nucleus was then connected to the WGA-positive vesicles and the angle of the lines was defined by ImageJ 1.54 f. The obtained data were imported into an Excel table. Because ImageJ 1.54 f defines angles in a range of $-180°$ to $+180°$, the data were converted into the range of 0°–360°, and the angles were normalized to 180°. Then we calculated the mean values for each cell and determined their deviation from 180°. This deviation was added to the angles of each corresponding cell to get the final dataset. If the obtained value was higher than 360°, we subtracted 360° from it, if it was lower than 0° it was corrected by the addition of 360°. Then we calculated the percentage of data in each bin for the whole datasets. We evaluated at least 15 cells of each cell type, and the data were copied to the Minitab (v.21) (https://www.minitab.com) followed by distribution analysis and histogram generation. Data were analyzed by the Levene's probe using the "Two sample variance test" for each combination of the groups as illustrated by a box plot diagram.

Western blots were analyzed by densitometry using the ImageJ software 1.54 f. Data from 3 to 4 experiments were statistically analyzed with One-Way ANOVA using the GraphPad Prism 8.0 software.

For *Homo sapiens* PMCA4b and *D. melanogaster* PMCA amino acid sequence comparison the P23634-6 and Q9V4C7 sequences were used from UniProt (UniProt, 2023). Alignment was depicted and analyzed by the Jalview 2.11.3.1 software[91] using the "Percentage Identity" dynamic color scheme. Conservation values were automatically calculated in Jalview as described in the paper of Livingstone and Barton[92].

The number of evaluated images or cells (n) and repetition numbers of experiments are indicated in figure legends.

## Reporting summary

Further information on research design is available in the Nature Portfolio Reporting Summary linked to this article.

## Data availability

Clinical data for 109 breast cancer patients (https://doi.org/10.6084/m9.figshare.26077291), original Western blots (https://doi.org/10.6084/m9.figshare.26068924), supplementary videos (https://doi.org/10.6084/m9.figshare.28360466) and raw data with the exact p-values in Excel format (https://doi.org/10.6084/m9.figshare.28202549) supporting the study's findings are deposited in Figshare. Other source data supporting the findings of this study are available from the corresponding author upon reasonable request.

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

## Acknowledgements

The authors wish to thank Mónika Truszka for her help with electron microscopy, and Dr. John T. Penniston for his continuous support. This research was supported by the National Research, Development and Innovation Office under grant numbers NRDI K135811 (to A.E.), K135757 (to A.T.) PD135447 (to T.C.) and TKP2021-EGA-24 (to A.E. and A.T.). The New National Excellence Program of the Ministry for Innovation and Technology from the source of the National Research, Development and Innovation Office (ÚNKP-23-5-ELTE-603 to T.C.) and the János Bolyai Research Scholarship of the Hungarian Academy of Sciences (BO/00023/21/8 to T.C.). The manuscript is accessible as a preprint on bioRxiv online preprint archive under the following link: https://www.biorxiv.org/content/10.1101/2024.01.20.576436v2.article-info#page.

## Author contributions

S.T. and A.E. designed the research and wrote the paper, S.T., D.K., J.S., R.N., and R.P. designed and performed experiments with human cell lines and analysed the data, A.M.T., A.E., and B.P. evaluated human tissue samples, A.E. analysed data available at the UALCAN web portal and the Kaplan–Meier Plotter online tool database, S.T. designed and performed Drosophila experiments, S.T., J.S., A.M.T., K.V., and A.E. performed the statistical analysis, J.S. and T.C. performed electron microscopy experiments, B.P. and A.T. reviewed and edited the manuscript, A.E. supervised the project. All authors read and approved the final version of the manuscript.

## Funding

## Competing interests

The authors declare no competing interests.

## Ethics

Clinicopathological data between 2000 and 2010 were obtained from the files of the Semmelweis University, 2nd Dept. of Pathology and from the Semmelweis University Health Care Database with the permission of the Hungarian Medical Research Council (ETT-TUKEB 14383/2017). Written informed consent was obtained from all patients.
