## [Transparent Peer Review file · Communications Biology]

The calcium pump PMCA4b promotes epithelial cell polarization and lumen formation

Corresponding Author: Dr Sarolta Tóth

Version 0:

Reviewer comments:

Reviewer #1

(Remarks to the Author)

The current study investigates the role of PMCA4b in epithelial cell polarization. The results indicate that PMCA4 is downregulated in tumor tissue samples collected from patients with luminal breast cancer as compared to non-tumoral tissue samples. PMCA4b is shown to be needed for the plasma membrane localization of E-cadherin in MCF-7 breast cancer cells and regulates cell polarity. In addition, the authors show that PMCA4b is required for pre-lumen and lumen formation in 2D and 3D, respectively, cultures of MCF7 cells, and regulates normal lumen morphology in larval salivary gland. Finally, the authors conclude that PMCA4b promotes the polarization of luminal epithelial cells. The study is interesting, timely and carefully performed and the interpretations are supported by the experimental data. The study is a nice mixture of basic and translational analysis using different cell types, well justified in tumor samples and a single luminal breast cancer cell line and progressing towards more general cell models.

Major points:

- In addition to patient samples, the studies in tumor cells are focused on a single cell line, it would be important to confirm some essential results in at least another luminal breast cancer cell line, i.e. the role of PMCA4b in pre-lumen or lumen formation/polarization.
- Is cell polarization regulated by calcium? i.e. is the calcium oscillation pattern altered when the expression of PMCA4b is modified in these conditions?

Minor:

The description of cell polarization in 2D or 3D cultures seems to be focused or explained in terms of an acinar structure but MCF7 cells, as well as the majority of breast cancer cells derive from ductal cells. Therefore, one would expect a tube formation instead of an acinar structure (with apical secretory vesicles).

Reviewer #2

(Remarks to the Author)

Enyedi and co-workers investigated the role of plasma membrane Ca²⁺ pump PMCA4 in regulating cell polarity and lumen formation of MCF-7 breast cancer cells and found that the ADP-ribosylation factor 6 (Arf6) small GTPase protein behaves as regulator of PMCA4b trafficking and that PMCA4b promotes polarization and lumen formation in 2D and 3D MCF-7 cells. In the light of these results the authors propose a correlation between the expression level of PMCA4, polarization and lumen formation and patient survival.

The experiments are well designed and the conclusion regarding PMCA4b trafficking and polarization promotion is justified by the results. However, I have some perplexity on the general significance of these findings and in extending the conclusion to an association between elevated PMCA4 expression with longer relapse free survival in patients and lower risk of cancer progression. Although Figure 1A shows PMCA4 downregulation in LUMA, LUMB1 and LUMB2 mammary tumors, instead panels on the right show an inverse correlation in the HER2-expressing LUMB2 subtype, in which poorer outcomes were associated with higher ATP2B4 expression (and even with ATP2B1).

Some specific points need to be addressed:

- Figure1 It is worth to check also PMCA2 levels in addition of PMCA1 being this isoform relevant for mammary gland.

Possible compensation by different isoforms could be relevant to better explain the correlation between PMCA expression and breast cancer.

- According to the description of Figure 2 in the text, PMCA4b downregulation affects E-cadherin distribution on the membrane and cell polarity. However, the quantification for E-cadherin and Dlg1 particles respectively shown in panel E and K is not intuitive: which is their link with fluorescence intensity profile? How was selected the position of the white line? It seems very arbitrary, and the information is rather qualitative than quantitative. What is the meaning of number of particles? Why should its increase correspond to reduced E-cadherin or Dlg1 expression on plasma membrane? It is not clear. And how can this calculation consider possible differences in the level of expression of E-cadherin or Dlg1?

The authors at lines 166-168 of page 6 reported that neither silencing nor over-expression of PMCA4b affected the expression levels of the epithelial and mesenchymal markers E-cadherin, N-cadherin and vimentin. This statement should be supported by a quantitative analysis by densitometry on bands intensity performed in at least n= 3 independent WB; instead the representative image of Panel A shows a reduced E-cadherin band intensity for sh-PMCA4 and GFP-PMCA4b or GFP-PMCA4bLA cells (suggesting loss of epithelial marker) and no signal at all for N-cadherin and vimentin was detected, which are supposed to increase during EMT but to be not expressed in MCF-7 epithelial cells.

The level of PMCA4 silencing or overexpression in the selected MCF-7 clones should be also quantified. Dlg1 also seems to be reduced in PMCA4 overexpressing clones in the WB and also from the fluorescence intensity profile. The author instead commented that it is unchanged (lines 188-189).

The analysis performed on Dlg1 in Panel H, I, J should be shown also for GFP_PMCA4LA transfected cells as done for E-cadherin

A clearer explanation of the meaning of number of E-cadherin or Dlg1 particles and a comment on fluorescence intensity profile should be done in the text to understand why these results indicate that PMCA downregulation affects E-cadherin distribution on the membrane and cell polarity.

-similar comment for Figure 3 showing ezrin distribution. What is the ratio of polarized cells (%) at the ordinate of Figure 3D? The authors should explain how these values have been obtained. Why the effect on GFP_PMCA4bLA clone is similar than in clone where PMCA4b was downregulated? Does the endocytotic defective PMCA4 act as dominant negative?

The paragraph at lines 169- 174 should be reconsidered:

However, silencing PMCA4 induced internalization of E-cadherin, resulting in its reduced plasma membrane localization. In contrast, the parental, GFP-PMCA4b- or the trafficking-mutant GFP-PMCA4bLA-expressing MCF-7 cells displayed more prominent plasma membrane localization of E-cadherin (Fig. 2B-E, Suppl. Fig. 2C). This suggests that loss of PMCA4 may induce partial epithelial-mesenchymal transition (EMT) through E-cadherin internalization without affecting the overall expression of the EMT markers.

There is no evidence for these conclusions: the statistical analysis of Figure 2B simply show that the number of E-cadherin particles is increased upon PMCA4 silencing and not changed upon PMCA4 overexpression. The authors' comment on this is not appropriate.

Similarly also the following affirmation should be reconsidered: no evidence for EMT.

Lines 173-174 This suggests that loss of PMCA4 may induce partial epithelial-mesenchymal transition (EMT) through E-cadherin internalization without affecting the overall expression of the EMT markers.

- lines 230-236: To inhibit Arf6 we treated PMCA4b-expressing MCF-7 230 cells with NAV2729 (Yoo et al., 2016). After 24 hours of the treatment, we observed an enrichment of PMCA4b-CD147 positive vesicles near the plasma membrane, and after 48 hours PMCA4b-CD147 vesicles accumulated in the cytoplasm (Fig. 5C) suggesting that Arf6 was involved in the endocytic recycling of the PMCA4b-CD147 complex.

These results should be validated by genetically ablating Arf6

- the data of Figure 5 are qualitative, a quantification should be performed

In summary, the paper is overall well written but needs to be substantially revised in the explanation of the data mentioned above and their interpretation upon performing the suggested control. Some of the conclusion should be taken with more caution.

Version 1:

Reviewer comments:

Reviewer #1

(Remarks to the Author)

The authors have adequately addressed the previous concerns and improve the manuscript.

Reviewer #2

(Remarks to the Author)

Dear Authors,

I appreciated very much the work you have done to improve the manuscript by deeply answering my points and adding new experiments to further corroborate your findings.

The manuscript nicely contributes to our knowledge on the signaling role of PMCA, that is well behind to keep cytosolic calcium concentration in resting conditions.

The finding that PMCA4 is implicated in promoting epithelial cell polarization and lumen formation, as well as the finding that its downregulation is associated with luminal A and B1 tumors, gave important molecular insights in the process of EMT in epithelial cells.

Dear Editor,

We greatly appreciate the reviewer's careful evaluation of our manuscript and raising insightful comments and important issues. Their feedback has been invaluable in improving the quality and clarity of our work. In response to the reviewers' suggestions, we have not only addressed all the points raised but also introduced new experiments to further strengthen our findings. These additions, along with the revisions made, aim to provide greater clarity and depth to the key results and implications of our study. We hope the revised version meets the reviewers' expectations and demonstrates our commitment to addressing satisfactorily the reviewers' constructive feedback.

Answers to Reviewers' comments:

Reviewer #1 (Remarks to the Author):

"The current study investigates the role of PMCA4b in epithelial cell polarization. The results indicate that PMCA4 is downregulated in tumor tissue samples collected from patients with luminal breast cancer as compared to non-tumoral tissue samples. PMCA4b is shown to be needed for the plasma membrane localization of E-cadherin in MCF-7 breast cancer cells and regulates cell polarity. In addition, the authors show that PMCA4b is required for pre-lumen and lumen formation in 2D and 3D, respectively, cultures of MCF7 cells, and regulates normal lumen morphology in larval salivary gland. Finally, the authors conclude that PMCA4b promotes the polarization of luminal epithelial cells. The study is interesting, timely and carefully performed and the interpretations are supported by the experimental data. The study is a nice mixture of basic and translational analysis using different cell types, well justified in tumor samples and a single luminal breast cancer cell line and progressing towards more general cell models."

Referee #1's comment	Reply
1.1. In addition to patient samples, the studies in tumor cells are focused on a single cell line, it would be important to confirm some essential results in at least another luminal breast cancer cell line, i.e. the role of	1.1. In response to the reviewer's request, we generated a new GFP-PMCA4b over-expressing cell line from another luminal-type breast cancer cell line, T47D. We found, that similarly to the MCF-7 model system, PMCA4b over-expression resulted in more pre-lumen-like structures in 2D cultures in GFP-PMCA4b-carrying T47D cells than in the parental T47D cells. These results confirm our data on the role of PMCA4b in pre-lumen formation by luminal breast cancer cells. The data are presented in Supplementary Fig. 4 (also shown below), and a short paragraph describing the results on T47D is included in lines 265-267* as follows: " PMCA4b overexpression elevated the number of pre-lumen-like structures in T47D luminal breast cancer cells, as well

* Lines indicates locations in the new manuscript with corrections.

PMCA4b in pre-lumen or lumen formation/polarization

(Supplementary Fig. 4A-C), further supporting the role of PMCA4b in lumen formation in luminal cell types.”

New Supplementary Fig. 4:

Title: “PMCA4b overexpression enhances pre-lumen formation in T47D cells.”

Figure legend: “**A** The Western blot shows expression of PMCA4 in parental and GFP-PMCA4b-expressing T47D cells. **B** Phalloidin staining of parental and GFP-PMCA4b-expressing T47D cell lines. White arrows point to pre-lumens. **C** Statistical analysis of pre-lumen formation in parental, and GFP-PMCA4b-expressing T47D cells. Graph displays means with 95% CI; data were collected from 2 independent experiments, $n_{(T47D)}=26$, $n_{(PMCA4b)}=31$. Data were analyzed with the Mann-Whitney test, p value: * $p<0.05$.”

Showing in these new experiments that PMCA4b is involved in cell polarization not only in MCF-7 cells and normal Drosophila salivary gland epithelium, but also in another breast cancer cell line (T47D), further underlines the validity of our observations.

1.2. Is cell polarization regulated by calcium? i.e. is the calcium oscillation pattern altered when the expression of PMCA4b is modified in these conditions?

1.2. We thank the reviewer for the question regarding the role of Ca^{2+} signaling in cell polarization and the potential influence of PMCA4b expression. Indeed, it is well-established that Ca^{2+} signaling plays a critical role in cell polarization, and that PMCA4 is essential in establishing the back-to-front Ca^{2+} gradient in migrating endothelial cells, which we referenced in our original Discussion section (lines 408-412).

As suggested by the reviewer, we performed a more extended experimental analysis of Ca^{2+} signaling in MCF-7 cells with varying PMCA4 expression. Our findings indicate that modifications in PMCA4b expression and/or trafficking significantly alter the Ca^{2+} signaling pattern in response to purinergic stimuli in these cells. Additionally, we discussed in the revised manuscript the findings of a recent study showing that purinergic stimulation of organoids derived

from murine breast cancer tissue results in a pronounced elevation of Ca^{2+} concentration levels when compared to organoids from normal tissue¹. This observation also aligns with our results in the MCF-7 cell model system, and further underscores the relationship between Ca^{2+} dynamics, PMCA4b expression and cancer cell behavior.

We have now incorporated these findings in the Results section of the revised manuscript (lines 205-222) as follows:

“PMCA4b is a critical mediator of Ca^{2+} signaling

Prolonged Ca^{2+} signaling has been shown to induce partial EMT in epithelial cells, characterized by the internalization of E-cadherin². Previous studies suggest that in breast epithelial cells, PMCA4 plays a critical role in removing Ca^{2+} after stimuli to terminate the Ca^{2+} signal³.⁴ Here, we demonstrate that purinergic stimuli induced by extracellular ATP resulted in a markedly elevated Ca^{2+} response in sh-PMCA4 and parental (PMCA4-low) cells when compared to cells overexpressing wild type PMCA4b or the trafficking mutant PMCA4b^{LA} (Fig. 4A-B). In cells with low PMCA4b abundance (sh-PMCA4 expressing and parental cells), the Ca^{2+} response exhibited two distinct peaks. The first peak likely corresponds to Ca^{2+} release from the endoplasmic reticulum (ER), while the second peak may involve Ca^{2+} entry through store-operated calcium channels (SOCs)⁵. In contrast, cells overexpressing PMCA4b^{LA} showed a rapid return to the baseline Ca^{2+} level after the first peak, with only minor fluctuations reminiscent of the second peak seen in sh-PMCA4 and parental cells, in good correlation with our previous work⁶. Cells overexpressing wild-type PMCA4b exhibited a significantly reduced Ca^{2+} response, as evidenced by a lower area under the curve and peak response to ATP (Fig. 4B-C). Our observations in MCF-7 cells suggest that the loss of PMCA4b leads to highly dysregulated Ca^{2+} signaling in breast cancer cells.”

These findings are discussed in the Discussion section of the revised manuscript (lines 356-362) as follows:

“Enhanced Ca^{2+} signaling was recently observed in organoids derived from breast cancer tissues in response to purinergic stimuli¹ that mirrors the patterns seen in MCF-7 cells with low PMCA4 abundance in our work. In contrast, the diminished Ca^{2+} signaling in MCF-7 cells with high PMCA4b abundance resembles that of organoids from normal tissue. This further supports the notion that PMCA4b can prevent sustained Ca^{2+} signals, which may otherwise promote EMT and contribute to the maintenance of the malignant phenotype.”

Methods section was supplemented with the following text on Ca^{2+} signaling in lines 544-555:

“ Ca^{2+} signal measurement

MCF-7 cells were seeded in 8-well chamber (155411, Nunc) and 48 hours later were transfected with CMV-R-GECO1 (a gift from Robert Campbell (32444, Addgene)⁷ using FuGENE HD transfection reagent (Promega Corporation), according to the manufacturer’s

recommendations. 48 hours after transfection, DMEM was replaced by Hanks's Buffered Salt Solution (HBSS, Thermo Fischer 88284) supplemented with 10 mM HEPES (pH 7.4) and 2 mM CaCl_2 . After 2 minutes baseline imaging, 10 μM ATP was added to the cells. Images were acquired every 0.4 seconds for 10 minutes. The relative fluorescence was calculated as F/F_0 (where F_0 was the average of 30 seconds baseline fluorescence). Time-lapse sequences were recorded with a Nikon Eclipse Ti2 confocal microscope with a Plan-Apochromat 20 \times /0.75 Air objective with iMSPECTOR software (Abberior Instruments GmbH) at room temperature." and Ca^{2+} signal analysis are described in lines 754-755: For Ca^{2+} signaling analysis the area under curve and first peak maximum values were calculated in GraphPad Prism 8.0." and in lines 754-755: "For Ca^{2+} signaling analysis the area under curve and first peak maximum values were calculated in GraphPad Prism 8.0."

Results are presented in a new figure, Fig. 4:

Title: "PMCA4b mediates purinergic receptor activated Ca^{2+} signaling in MCF-7 cells."

Figure legend: "A Intracellular Ca^{2+} was measured with the R-GECO fluorescence indicator in parental, PMCA4-specific shRNA (sh-PMCA4), GFP-PMCA4b and GFP-PMCA4b^{LA}-expressing MCF-7 cells. 10 μM ATP was added at the 120 second mark. Signals from individual cells are illustrated using different colored lines. Number of analyzed cells: $n_{(\text{MCF-7})}=12$, $n_{(\text{sh-PMCA4})}=15$, $n_{(\text{PMCA4b})}=18$, $n_{(\text{PMCA4bLA})}=14$. Data are derived from two independent experiments. B Area under curve (AUC) values were calculated from curves in panel A and analyzed with Kruskal-Wallis and Dunn's multiple comparisons tests; adjusted p values: ** $p<0.01$, *** $p<0.001$; non-significant difference is not labeled. Graph displays means with 95% CI. C First peak F/F₀ maximum values were calculated from data in panel A and analyzed with Kruskal-Wallis and Dunn's multiple comparisons tests; adjusted p values: ** $p<0.01$, *** $p<0.001$; non-significant difference is not labeled."

1.3. The description of cell polarization in 2D or 3D cultures seems to be focused or explained in terms of an acinar structure but MCF7 cells, as well as the majority of breast cancer cells derive from ductal cells. Therefore, one would expect a tube formation instead of an acinar structure (with apical secretory vesicles).

1.3. We agree with the reviewer's comment, and we have now avoided designating our luminal structures as acini in the revised manuscript. We thank the reviewer for pointing this out, as at this stage it is not possible to distinguish between ductal and acinar lumens without the use of further specific markers. Interestingly, a recent study demonstrated that MCF-7 and T47D cells stably expressing integrin $\alpha\beta3$ can differentiate into well-organized "acinar-like" structures both *in vitro* and *in vivo*⁸. We agree with the reviewer that it is not correct to call these structures acini. This notwithstanding, the findings in a previous article⁸ indicate that these cell lines can form lumens under certain conditions. We now discuss the results of this study in the Discussion section of our manuscript (lines 381-388) to provide additional context as follows:

"In a previous paper, the authors reported that MCF-7 and T47D luminal A-type breast cancer cells stably expressing Int- $\alpha\beta3$ differentiated into well-organized "acinar-like" structures both in vitro and in vivo⁸, resembling the central lumen-forming mammospheres generated by our PMCA4b-expressing MCF-7 cells. We have demonstrated that introducing PMCA4b to a BRAF mutant melanoma cell line resulted in a severe loss of integrin $\beta4$ expression⁹. These findings suggest that PMCA4 may play a role in lumen-formation by regulating integrin-mediated cell adhesion, highlighting the need for further investigation in this respect."

In response to the reviewer's comment, we replaced the term "secretory vesicles" with "intracellular vesicles" in line 290. Additionally, at the end of the paragraph, we revised the original sentence to state in lines 297-300: *"These findings further support the importance of PMCA4b as an essential regulator of vesicular trafficking and its possible role in secretion."* This replaces the previous sentence (lines 295-298): *"Moreover, in PMCA4b^{LA}-expressing cells, small endocytic vesicles accumulated near the plasma membrane (Fig. 8D), underlining the importance of endosomal trafficking in secretion."*

Furthermore, in the last paragraph of the Results section (lines 320-321), we have removed statements about secretion in MCF-7 cells, such as the comparison to PMCA4-silenced MCF-7 mammospheres (the removed sentence: "similar to that seen in the PMCA4-silenced MCF-7 mammospheres"). Instead, the last sentence of the Discussion states: *"These results highlight the evolutionarily conserved role of PMCA in the regulation of lumen morphology, and more specifically merocrine secretion in the Drosophila larval salivary gland."*

Similarly, in the Discussion (lines 394-397), we now address secretion by referencing the Drosophila results only: *"Our electron microscopy analysis of PMCA-deficient Drosophila salivary glands revealed a pronounced secretion defect and abnormal accumulation of secretory vesicles and in vivo (Fig 9G)."*

Reviewer #2 (Remarks to the Author):

“Enyedi and co-workers investigated the role of plasma membrane Ca²⁺ pump PMCA4 in regulating cell polarity and lumen formation of MCF-7 breast cancer cells and found that the ADP-ribosylation factor 6 (Arf6) small GTPase protein behaves as regulator of PMCA4b trafficking and that PMCA4b promotes polarization and lumen formation in 2D and 3D MCF-7 cells. In the light of these results the authors propose a correlation between the expression level of PMCA4, polarization and lumen formation and patient survival.

The experiments are well designed and the conclusion regarding PMCA4b trafficking and polarization promotion is justified by the results. However, I have some perplexity on the general significance of these findings and in extending the conclusion to an association between elevated PMCA4 expression with longer relapse free survival in patients and lower risk of cancer progression.”

Referee #2's comment	Reply
2.1. Although Figure 1A shows PMCA4 downregulation in LUMA, LUMB1 and LUMB2 mammary tumors, instead panels on the right show an inverse correlation in the HER2-expressing LUMB2 subtype, in which poorer outcomes were associated with higher ATP2B4 expression (and even with ATP2B1).	2.1. We agree with the reviewer that the inverse correlation between the breast cancer sub-types is quite surprising although not unique, since E-cadherin and Scrib also can have opposing functions in different tumor types. This is now discussed in the Discussion section of the present manuscript (lines 442-444)[†]: “This is not unexpected, since “tumor suppressors” like E-cadherin and Scrib may exhibit opposing functions across different tumor types, cancer stages, and/or at metastatic sites^{10, 11}”. The outlier status of HER2-positive tumors may be related also to the direct effect of HER2 exerted on cell adhesion and migration as reported recently¹² that sets HER2-positive (such as LUMB2) tumors apart from LUMA A and LUMB1 in terms of patient survival.
2.2. Some specific points need to be addressed: -Figure1 It is worth to check also PMCA2 levels in addition of PMCA1 being this isoform relevant for mammary gland.	2.2. We appreciate the reviewer’s comment, and we thus investigated the expression profile of PMCA2 at the mRNA level in luminal breast cancer cases, comparing it to normal breast tissue, using data from The Cancer Genome Atlas (TCGA). The results are added to the first section of the Results in lines 135-138, as follows: “Importantly, the mRNA level of PMCA2 was much lower in normal non-lactating breast tissue compared to the other isoforms, and no significant

[†] Lines indicates locations in the new manuscript with corrections.

Possible compensation by different isoforms could be relevant to better explain the correlation between PMCA expression and breast cancer.

difference was observed between normal tissue and luminal breast cancer cohorts (Supplementary Fig.1A).”

This is also discussed in the Discussion section in lines 335-338: “It is important to note that, in contrast to the elevated levels of PMCA2 reported in HER2-positive and basal-type tumors^{13, 14}, PMCA2 expression remains low in luminal breast tumor subtypes according to data from the CPTAC-GWAS databases (Supplementary Fig. 1A).”

Since no protein data are currently available for PMCA2, for comparison we also examined the mRNA expression levels of the other two PMCA isoforms and have included these data in Supplementary Fig. 1A, as also shown below:

Title:

Title: “PMCA expression in normal and luminal type breast cancer patients”

Figure legend: “A mRNA level comparison of PMCA1, PMCA2 and PMCA4 in normal breast tissue and luminal breast cancer subtypes. Data derived from The Cancer Genome Atlas, TCGA of the Clinical Proteomic Tumor Analysis Consortium (CPTAC) (<http://ualcan.path.uab.edu>) database and analyzed by Student’s t test; $n_{(normal)}=114$, $n_{(luminal)}=566$; *** $p<0.001$; non-significant differences are not indicated.”

We included these changes in the first paragraph of the Results as follows in lines 129-135 (the changes are shown in bold): “Data from the Clinical Proteomic Tumor Analysis Consortium¹⁵ (CPTAC) (<https://ualcan.path.uab.edu>) confirmed these findings and showed significantly reduced PMCA4 **both at the mRNA (The Cancer Genome Atlas, TCGA) and protein levels in luminal breast cancer types compared to normal breast tissue. In contrast to PMCA4, PMCA1 expression was reduced at the mRNA level while no significant difference could be detected at the protein level in the same comparison** (Fig. 1C, Supplementary Fig. 1A).” We also noted this in the Discussion (lines 332-333): “These observations are in good correlation with CPTAC **protein and mRNA expression levels** of luminal breast cancer cases”.

Additionally, we added a sentence in the Introduction (lines 86-88) describing previous findings on PMCA2: “In contrast, PMCA2 mRNA expression was upregulated in HER2-positive- and basal-type breast cancers¹³, and its expression correlated with HER2 levels in HER2-positive tumors¹⁴.”

These revisions aim to enhance the clarity and completeness of the manuscript in addressing PMCA2 expression patterns in accordance with the reviewer.

2.3. According to the description of Figure 2 in the text, PMCA4b downregulation affects E-cadherin distribution on the membrane and cell polarity. However, the quantification for E-cadherin and Dlg1 particles respectively shown in panel E and K is not intuitive: which is their link with fluorescence intensity profile? How was selected the position of the white line? It seems very arbitrary, and the information is rather qualitative than quantitative. What is the meaning of number of particles?

2.3. The curves in Figure 2 show a qualitative intensity distribution of E-cadherin (Fig. 2D), Dlg1 (Fig. 2J) and GFP-PMCA4 fluorescence signals across the white line of the particular cells shown in B.

The legends to Figure 2D and 2J are corrected accordingly: “D: *Fluorescence intensity profiles of E-cadherin and GFP-PMCA4b corresponding to the white line across the cells shown in B.*” and “J: *Fluorescence intensity profiles of Dlg1 and GFP-PMCA4b corresponding to the white line across the cells shown on panel H.*” Methods section was supplemented with the following text (lines 744-750): “*Intensity profiles for E-cadherin and Dlg1 image analysis were created by using the ZEN 2 Software for obtaining intensity data and GraphPad Prism 8.0 for creating the graphs. Intensity was measured along a line that crosses the plasma membrane of two cells and avoids nuclei. Outlines, representing E-cadherin and Dlg1-positive cell compartments were created using ImageJ 1.54f by setting the auto threshold command and using Analyze Particles function to display bare outlines.*”

While the intensity curves described above are qualitative representation of E-cadherin and Dlg1 distribution across individual cells, Figures 2E and 2K correspond to quantitative evaluations of the number of intracellular vesicles (designated as particles in the original manuscript) per cell in the different cell lines. The legends to Figures 2E and 2K are corrected accordingly: “*Graph displays the means of the number of intracellular vesicles per cell with 95% confidence intervals (CI).*”

In case of Dlg1 we made a mistake in the original manuscript where the number of vesicles was not normalized by the number of cells, which is now corrected, and the number of intracellular vesicles (designated as particles in the original manuscript) per cell is determined similar to that described for E-cadherin in Figures 2E and 2K. The corrected graph is shown in Figure 2K and also inserted below.

Name of y-axis is changed from “*Number of E-cadherin particles*” to “*Number of E-cadherin vesicles/cell*” in Fig. 2E:

Improved figure legend (the changes are shown in bold): “**E** *Statistical analysis of the number of E-cadherin-positive vesicles in parental, PMCA4-specific shRNA (sh-PMCA4), GFP-PMCA4b and GFP-PMCA4b^{LA}-expressing MCF-7 cells. Graph displays the means of the number of intracellular vesicles per cell with 95% confidence intervals (CI); data were collected from 2 independent experiments, $n_{(MCF-7)}=13$, $n_{(sh-PMCA4)}=17$,*

$n_{(PMCA4b)}=13$, $n_{(PMCA4b^{LA})}=16$. Data were analyzed with Kruskal-Wallis and Dunn's multiple comparisons tests, adjusted p values: $**p<0.01$, $***p<0.001$. Non-significant differences are not indicated."

Name of y-axis is changed from "Number of Dlg1 particles" to "Number of Dlg1 vesicles/cell" and PMCA4b^{LA} data were added in Fig. 2K:

Improved figure legend (the changes are shown in bold): "K Statistical analysis of the number of Dlg1-positive particles in parental, PMCA4-specific shRNA (sh-PMCA4) and GFP-PMCA4b-expressing MCF-7 cells. Graph displays the means of the number of intracellular vesicles per cell with 95% CI; data were collected from 3 independent experiments, $n_{(MCF-7)}=14$, $n_{(sh-PMCA4)}=15$, $n_{(PMCA4b)}=17$, $n_{(PMCA4b^{LA})}=14$; data were analyzed with Kruskal-Wallis and Dunn's multiple comparisons tests; adjusted p values:

$***p<0.001$; non-significant difference is not labeled."

We thank the reviewer for these remarks.

2.4. Why should its increase correspond to reduced E-cadherin or Dlg1 expression on plasma membrane? It is not clear. And how can this calculation consider possible differences in the level of expression of E-cadherin or Dlg1?

2.4. The expression level of E-cadherin and Dlg1 was not modified by PMCA4 silencing; therefore, we suggest that the enhanced PMCA4 internalization - characterized by the number of intracellular vesicles – is proportional to its reduced plasma membrane localization. Since we did not measure directly the level of E-cadherin in the plasma membrane, we corrected the title of the second paragraph in the Results (lines 152-153): "PMCA4b silencing induced internalization of E-cadherin in MCF-7 breast cancer cells", and also in the corresponding sentence in the Results accordingly (lines 167-170): "Although silencing PMCA4 did not affect significantly the overall expression level of E-cadherin (Supplementary Fig. 2B), it did result in its internalization and localization in intracellular compartments (Fig. 2B-E)."

2.5. The authors at lines 166-168 of page 6 reported that neither silencing nor over-expression of PMCA4b affected the expression levels of the epithelial and mesenchymal markers E-cadherin, N-cadherin and vimentin. This statement should be supported by a quantitative analysis by densitometry on bands

2.5. In response to the reviewer's request, we performed quantitative analysis using densitometry of WB band's intensity of $n=3-4$ independent determinations. Indeed, no positive signals could be detected for N-cadherin and vimentin, and no significant differences in the expression of E-cadherin or Dlg1 were observed, suggesting the lack of a full/complete epithelial-mesenchymal transition in these cell lines. However, PMCA4 silencing induced E-cadherin and Dlg1 internalization, as also discussed above.

Western blot analyses of the cell lines are shown in Supplementary Figure 2B that is also inserted below.

intensity performed in at least n= 3 independent WB; instead the representative image of Panel A shows a reduced E-cadherin band intensity for sh-PMCA4 and GFP-PMCA4b or GFP-PMCA4b-LA cells (suggesting loss of epithelial marker) and no signal at all for N-cadherin and vimentin was detected, which are supposed to increase during EMT but to be not expressed in MCF-7 epithelial cells.

Added text to the Supplementary Figure 2 legend: “**B** Densitometry analysis of Western blot experiments for PMCA4, pan-PMCA, E-cadherin, N-cadherin, vimentin and Dlg1 proteins in MCF-7 and A375 cells, with the latter serving as a positive control. Graphs display means with standard deviation; data were collected from 3 or 4 independent experiments. Data were analyzed with ordinary one-way ANOVA and Tukey's multiple comparisons tests in the case of PMCA4, pan-PMCA, E-cadherin and Dlg1; adjusted p values: *p<0.05; **p<0.01; non-significant differences are not labeled. The Western blots did show positivity for N-Cadherin and Vimentin.”

2.6. The level of PMCA4 silencing or overexpression in the selected MCF-7 clones should be also quantified.

2.6. We quantified the protein levels of PMCA4 in the different cell lines, as requested. Data are shown in Supplementary Fig. 2 that is inserted above, under reply 2.5.

2.7. Dig1 also seems to be reduced in PMCA4 overexpressing clones in the WB and also from the fluorescence intensity profile. The author instead

2.7. This has now been also quantified, and no significant differences were observed between the cell lines. Data are shown in Supplementary Figure 2B that is inserted above under reply 2.5.

commented that it is unchanged (lines 188-189).	
2.8. The analysis performed on Dlg1 in Panel H, I, J should be shown also for GFP_PMCA4LA transfected cells as done for E-cadherin.	2.8. Quantitative analysis has been performed on Dlg1 for GFP-PMCA4b^{LA} cells according to the reviewer's suggestion. The results are now shown in the diagram of Fig 2K (see under reply 2.3.) and in Supplementary Fig. 2G, H, I (see below).  Added text to the Supplementary Figure 2 legend: “G Dlg1 immunostaining of GFP-PMCA4b^{LA}-expressing MCF-7 cells. H Outline of the Dlg1-positive cell compartments generated with the Image J software. I Fluorescence intensity profiles of Dlg1 and GFP-PMCA4b^{LA} across the line shown in G.”
2.9. A clearer explanation of the meaning of number of E-cadherin or Dlg1 particles and a comment on fluorescence intensity profile should be done in the text to understand why these results indicate that PMCA downregulation affects E-cadherin distribution on the membrane and cell polarity.	2.9. In response to the reviewer's comments, we clarified these issues on E-cadherin and Dlg1 internalization and distribution in the different cell lines through replies 2.2 to 2.7 above, and we hope that those satisfies the reviewer's criticism.

2.10. -similar comment for Figure 3 showing ezrin distribution. What is the ratio of polarized cells (%) at the ordinate of Figure 3D? The authors should explain how these values have been obtained.

2.10. This has been corrected as requested by the reviewer; the ordinate is now labelled as “percent of polarized cells” in Fig 3D (see below).

D

In the Methods we described how these values were obtained: “To define the percentage of polarized MCF-7 cells in 2D cell cultures, the number of cells with asymmetric and symmetric distribution of the apical marker ezrin was quantified manually. The cells were sorted based on their ezrin distribution into polarized and non-polarized groups.”

2.11. Why the effect on GFP_PMCA4b-LA clone is similar than in clone where PMCA4b was downregulated? Does the endocytotic defective PMCA4 act as dominant negative?

2.11. We greatly appreciate the reviewer’s insightful comment. However, addressing this question would require additional experiments, which are beyond the scope of the current study. We therefore plan to explore this interesting topic in future work, and we believe this issue does not modify the conclusions of the present manuscript.

2.12. The paragraph at lines 169- 174 should be reconsidered: However, silencing PMCA4 induced internalization of E-cadherin, resulting in its reduced plasma membrane localization. In contrast, the parental, GFP-PMCA4b- or the trafficking-mutant GFP-PMCA4b-LA-expressing MCF-7 cells displayed more prominent plasma membrane localization of E-cadherin (Fig. 2B-E, Suppl. Fig. 2C). This suggests that loss of PMCA4 may induce

2.12. According to the reviewer’s request we changed these sentences in the Results in section 2, in lines 167-170 as follows: “Although silencing PMCA4 did not affect significantly the overall expression level of E-cadherin (Suppl. Fig. 2B), it did result in its internalization and localization in intracellular compartments (Fig. 2B-E).” And in lines 173-176 as follows: “This suggests that loss of PMCA4 may induce partial epithelial-mesenchymal transition (EMT) through E-cadherin internalization, without significantly affecting the overall expression of E-cadherin and other EMT markers, similarly to published data (Aiello et al., 2018).” Also, to clarify the issue on the relationship of PMCA4 loss and partial EMT we introduced two additional paragraphs in the Discussion section in lines 345-366: “Partial EMT is a metastable hybrid state of tumor cells that exists between fully epithelial and fully mesenchymal characteristics, representing a distinct subpopulation within the tumor¹⁶. Recent findings suggest that partial EMT can involve the internalization of epithelial proteins, such as E-cadherin, rather than their transcriptional suppression¹⁷. This transient relocalization of E-cadherin in endocytic vesicles allows for the dynamic modulation of cell-cell adhesion, contributing to tumor cell invasiveness by enabling the re-establishment of cell-cell contacts during collective cell migration.

partial epithelial-mesenchymal transition (EMT) through E-cadherin internalization without affecting the overall expression of the EMT markers. There is no evidence for these conclusions: the statistical analysis of Figure 2B simply show that the number of E-cadherin particles is increased upon PMCA4 silencing and not changed upon PMCA4 overexpression. The authors' comment on this is not appropriate. Similarly, also the following affirmation should be reconsidered: no evidence for EMT. Lines 173-174 This suggests that loss of PMCA4 may induce partial epithelial-mesenchymal transition (EMT) through E-cadherin internalization without affecting the overall expression of the EMT markers.

*Interestingly, prolonged Ca²⁺ signaling has been implicated in partial EMT through its role in E-cadherin internalization². Notably, low levels of PMCA4 protein, observed in various tumor types (including luminal breast cancer cells and tissues) are associated with prolonged Ca²⁺ signaling^{3, 4, 18}, and may contribute to partial EMT. Enhanced Ca²⁺ signaling was recently observed in organoids derived from breast cancer tissues¹, and this mirrors the patterns seen in MCF-7 cells with low-PMCA4 abundance in our work. In contrast, the diminished Ca²⁺ signaling in MCF-7 cells with high PMCA4b abundance resembles that of organoids from normal tissue. This supports the notion that PMCA4b can prevent sustained Ca²⁺ signals, which may otherwise promote EMT and contribute to the maintenance of the malignant phenotype. Indeed, our findings demonstrate that PMCA4b is essential for the plasma membrane localization of E-cadherin and Dlg1 in MCF-7 cells (Fig 2, 3). Elevated PMCA4b levels also increase the number of polarized cells and promote polarized endosomal vesicular trafficking (Fig. 3, 4)”. In addition, minor changes are also made in the rest of the same paragraph as follows in lines 366-372 (changes are labelled in bold): “**These results suggest that PMCA4b enhances cell polarization, possibly through its interaction with the PDZ polarity protein Dlg1 (Fig. 2, 9). Interestingly, the loss of Dlg1 has been implicated in cancer progression across various tumor types highlighting the importance of its proper expression and/or localization** ¹⁹⁻²¹. **Notably, in contrast to our findings in MCF-7 cells, PMCA4 silencing in gastric cancer cells induces full EMT, suggesting cancer tissue-specific function of PMCA4** ²².”*

2.13. - lines 230-236: To inhibit Arf6 we treated PMCA4b-expressing MCF-7 cells with NAV2729 (Yoo et al., 2016). After 24 hours of the treatment, we observed an enrichment of PMCA4b-CD147 positive vesicles near the plasma membrane, and after 48 hours PMCA4b-CD147

2.13. We thank the reviewer for the insightful comment suggesting that our results should be validated by genetically ablating *Arf6*. We agree that confirming our findings using genetic approaches is an important step. To address this, we attempted to silence *Arf6* expression using two independent siRNA constructs (Sigma-Aldrich human ARF6 targeted MISSION esiRNA: cat. number: EHU137641; Thermo Fisher Human ARF6 silencer select siRNA: cat. number: 4390824) and two different transfection reagents (Rorche X-tremeGENE™ HP DNA transfection reagent; PolyPlus INTERFERin® siRNA and miRNA transfection reagent). Despite our efforts, these attempts were unsuccessful, as we were unable to achieve sufficient knockdown of *Arf6* to perform the proposed validation. Instead, we used a pharmacological inhibitor targeting the *Arf6* pathway, and transfected dominant-negative *Arf6* constructs to impair *Arf6* function; these have

vesicles accumulated in the cytoplasm (Fig. 5C) suggesting that Arf6 was involved in the endocytic recycling of the PMCA4b-CD147 complex.

These results should be validated by genetically ablating Arf6.

already been included in the original manuscript. We believe that this double approach served as a solid complementary validation of our findings. Given the technical challenges we encountered with siRNA knockdown, we respectfully ask the reviewer's understanding regarding the limitations of this method in our experimental context.

2.14. - the data of Figure 5 are qualitative, a quantification should be performed.

2.14. Thank you for this important critical remark. Quantitative analysis has been performed and is presented in Figure 6, panels D and E. The following text was added to the Methods section related to panel E (lines 756-761): *“For analyzing cytoplasmic PMCA4b localization after NAV2729 treatment the total and the cytoplasmic GFP-PMCA4b fluorescence signal intensities of individual cells were determined using the ImageJ software 1.54f (Supplementary figure 3). The ratio of cytoplasmic signal was calculated by dividing the cytoplasmic signal with the total signal.”* The following text was added to the Methods section related to panel D (lines 772-775): *“To analyze changes in the percentage of cells with internalized GFP-PMCA4b in response to NAV2729 treatment, cells were sorted into groups based on their PMCA4b localization (plasma membrane or vesicular) manually. From both analyses the number of cells was collected into contingency tables and data were analyzed by chi-square test.”*

Statistical analyses in Panels D and E were added to Figure 6, (Fig. 5 in the original manuscript):

Added text to the figure legends: **D** Statistical analysis of the distribution of cells with internalized GFP-PMCA4b in control and after 48 hours of NAV2729 treatment. Data were collected from 8 fields of view from 3 independent experiments; $n_{(PMCA4b\ control)}=589$, $n_{(PMCA4b + 48h\ NAV2729)}=435$. Data were analyzed with

*chi-square test, p value: *** $p < 0.001$; non-significant difference is not labeled.* **E** Statistical analysis of the GFP-PMCA4b signal distribution between cytoplasm and plasma membrane. Graph displays means with 95% CI; data were collected from 3 independent experiments, $n_{(PMCA4b\ control)}=40$, $n_{(PMCA4b +$

48h NAV2729)=40 cells, and analyzed as described in the Methods section. Data were analyzed with unpaired t test, p value: *** $p < 0.001$.”

In Fig. 6C, upper panel the original “- NAV2729” label was replaced by “control” label:

We have also included a new supplementary figure (Supplementary Fig. 3) to demonstrate the methods employed for this quantification (see below).

Title: “Determination of the relative cytosolic GFP-PMCA4b fluorescence signal in control, and NAV2729-treated MCF-7 cells.”
 Figure legend: “Images show the GFP-PMCA4b fluorescence signal in an individual control and NAV2729-treated

PMCA4b-expressing MCF7 cell. Yellow lines demonstrate areas where intensity was measured for Fig. 6 panel E. The outer line encircles total cell area and the inner line encircles the cytoplasmic area. The proportion of the cytoplasmic PMCA4b signal was calculated by dividing the cytoplasmic signal with the total signal.”

“In summary, the paper is overall well written but needs to be substantially revised in the explanation of the data mentioned above and their interpretation upon performing the suggested control. Some of the conclusion should be taken with more caution.”

We thank the remarks of the reviewers, these improved our work. We believe that we answered satisfactorily to the issues raised, and hope that our manuscript is now acceptable for publication.

Sincerely yours,

Sarolta Tóth and Agnes Enyedi

Additonal changes

During the revision period we realized a few deficiencies in the manuscript that we corrected in the text or in the figures. It is important to note that neither of the changes listed below affects the conclusions of the present manuscript.

1. Exact origin of HEK293 cells was indicated in Methods section (lines 510-511): „...HEK-293T (**cat. number: CRL-1573**) cell lines were purchased from the American Type Culture Collection (ATCC).” The change is shown in bold.

2. In Supplementary Fig. 2A the N-Cadherin and Vimentin blot slides were swapped. We apologize for this error, which is now corrected.

3. The manuscript has been supplemented with Data availability (lines 802-808) and Inclusion and ethics declarations (lines 822-829) sections:

„Data availability

Clinical data for 109 breast cancer patients (<https://doi.org/10.6084/m9.figshare.26077291>), original Western blots (<https://doi.org/10.6084/m9.figshare.26068924>) and raw data with the exact p-values in Excel format (<https://doi.org/10.6084/m9.figshare.28202549>) supporting the study's findings are deposited in Figshare. Other source data supporting the findings of this study are available from the corresponding author upon reasonable request.”

“Inclusion and ethics

Competing interests

The authors declare no competing interests.

Ethics

Clinicopathological data between 2000 and 2010 were obtained from the files of the Semmelweis University, 2nd Dept. of Pathology and from the Semmelweis University Health Care Database with the permission of the Hungarian Medical Research Council (ETT-TUKEB 14383/2017). Written informed consent was obtained from all patients.

4. A minor mistake was corrected in Fig. 1B.

5. Minor mistakes were corrected in Supplementary Table 1. Affected part of the original and corrected version are included below.

Subtypes	LUMA	30	9	39	0.142
	LUMB1	41	9	50	
	LUMB2	12	8	20	
	Total	83	26	109	
Age	Mean	57.54	61.27		0.186
	SD	12.59	12		
	Range	27- 80	44- 83		
	Total	83	26	109	
Grade	1	23	6	29	0.519
	2	41	10	51	
	3	18	8	26	
	Total	82	24	106	

pT	1	49	13	62	0.492
	2	33	12	45	
	Total	82	25	107	
pN	0	49	14	63	0.923
	1	33	9	42	
	Total	82	23	105	
Metastatic	Yes	13	3	16	0.621
	No	68	22	90	
	Total	81	25	106	

Original

Subtypes	LUMA	30	9	39	0.1066
	LUMB1	41	9	50	
	LUMB2	12	8	20	
	Total	83	26	109	
Age	Mean	57.33	61.96		0.1615
	SD	12.4	12.4		
	Range	27- 80	35- 83		
	Total	83	26	109	
Grade	1	24	5	29	0.5789
	2	39	12	51	

	3	19	7	26	
	Total	82	24	106	
pT	1	49	13	62	0.4143
	2	33	12	45	
	Total	82	25	107	
pN	0	49	14	63	0.5142
	1	32	10	42	
	Total	82	23	105	
Metastatic	Yes	12	4	16	0.958
	No	69	21	90	
	Total	81	25	106	

Corrected

References

1. Henningsen, M.B. *et al.* Amplified Ca(2+) dynamics and accelerated cell proliferation in breast cancer tissue during purinergic stimulation. *Int J Cancer* **151**, 1150-1165 (2022).
2. Norgard, R.J. *et al.* Calcium signaling induces a partial EMT. *EMBO Rep* **22**, e51872 (2021).
3. Varga, K. *et al.* Histone deacetylase inhibitor- and PMA-induced upregulation of PMCA4b enhances Ca2+ clearance from MCF-7 breast cancer cells. *Cell Calcium* **55**, 78-92 (2014).
4. Padanyi, R. *et al.* Multifaceted plasma membrane Ca(2+) pumps: From structure to intracellular Ca(2+) handling and cancer. *Biochim Biophys Acta* **1863**, 1351-1363 (2016).
5. Wang, L., Roger, S., Yang, X.B. & Jiang, L.H. Role of the store-operated Ca2+ channel in ATP-induced Ca2+ signalling in mesenchymal stem cells and regulation of cell functions. *Front Biosci (Landmark Ed)* **26**, 1737-1745 (2021).
6. Paszty, K. *et al.* Plasma membrane Ca(2+)-ATPases can shape the pattern of Ca(2+) transients induced by store-operated Ca(2+) entry. *Sci Signal* **8**, ra19 (2015).
7. Zhao, Y. *et al.* An expanded palette of genetically encoded Ca(2+) indicators. *Science* **333**, 1888-1891 (2011).
8. Abu-Tayeh, H. *et al.* 'Normalizing' the malignant phenotype of luminal breast cancer cells via alpha(v)beta(3)-integrin. *Cell Death Dis* **7**, e2491 (2016).

9. Naffa, R. *et al.* P38 MAPK Promotes Migration and Metastatic Activity of BRAF Mutant Melanoma Cells by Inducing Degradation of PMCA4b. *Cells* **9** (2020).
10. Kowalski, P.J., Rubin, M.A. & Kleer, C.G. E-cadherin expression in primary carcinomas of the breast and its distant metastases. *Breast Cancer Res* **5**, R217-222 (2003).
11. Peglion, F. & Etienne-Manneville, S. Cell polarity changes in cancer initiation and progression. *J Cell Biol* **223** (2024).
12. Wang, G. *et al.* Her2 promotes early dissemination of breast cancer by inhibiting the p38 pathway through the downregulation of MAP3K4. *Cell Commun Signal* **22**, 611 (2024).
13. Peters, A.A. *et al.* The calcium pump plasma membrane Ca(2+)-ATPase 2 (PMCA2) regulates breast cancer cell proliferation and sensitivity to doxorubicin. *Sci Rep* **6**, 25505 (2016).
14. Jeong, J. *et al.* PMCA2 regulates HER2 protein kinase localization and signaling and promotes HER2-mediated breast cancer. *Proc Natl Acad Sci U S A* **113**, E282-290 (2016).
15. Chandrashekar, D.S. *et al.* UALCAN: An update to the integrated cancer data analysis platform. *Neoplasia* **25**, 18-27 (2022).
16. Pal, A., Barrett, T.F., Paolini, R., Parikh, A. & Puram, S.V. Partial EMT in head and neck cancer biology: a spectrum instead of a switch. *Oncogene* **40**, 5049-5065 (2021).
17. Aiello, N.M. *et al.* EMT Subtype Influences Epithelial Plasticity and Mode of Cell Migration. *Dev Cell* **45**, 681-695 e684 (2018).
18. Hegedus, L. *et al.* Histone Deacetylase Inhibitor Treatment Increases the Expression of the Plasma Membrane Ca(2+) Pump PMCA4b and Inhibits the Migration of Melanoma Cells Independent of ERK. *Front Oncol* **7**, 95 (2017).
19. Catterall, R., Lelarge, V. & McCaffrey, L. Genetic alterations of epithelial polarity genes are associated with loss of polarity in invasive breast cancer. *Int J Cancer* **146**, 1578-1591 (2020).
20. Fuja, T.J., Lin, F., Osann, K.E. & Bryant, P.J. Somatic mutations and altered expression of the candidate tumor suppressors CSNK1 epsilon, DLG1, and EDD/hHYD in mammary ductal carcinoma. *Cancer Res* **64**, 942-951 (2004).
21. Marziali, F., Dizanzo, M.P., Cavatorta, A.L. & Gardiol, D. Differential expression of DLG1 as a common trait in different human diseases: an encouraging issue in molecular pathology. *Biol Chem* **400**, 699-710 (2019).
22. Wang, T. *et al.* The calcium pump PMCA4 prevents epithelial-mesenchymal transition by inhibiting NFATc1-ZEB1 pathway in gastric cancer. *Biochim Biophys Acta Mol Cell Res* **1867**, 118833 (2020).

Dario Ummarino, PhD
Senior Editor
Communications Biology

Kaliya Georgieva, PhD
Associate Editor
Communications Biology

February 14th 2025

Response to Reviewers

Manuscript ID: COMMSBIO-24-3274-T

Dear Editors,

We appreciate the constructive feedback provided by the reviewers and their positive assessment of our revised manuscript. We are pleased that both reviewers acknowledge the improvements made and the additional experimental evidence provided.

Reviewer 1: We are grateful for the reviewer's positive comments and are pleased that our revisions adequately addressed the previous concerns.

Reviewer 2: We greatly appreciate the reviewer's thoughtful feedback and kind remarks on the significance of our findings.

We thank both reviewers for their time and effort in evaluating our work. We hope that the revised manuscript meets the journal's standards for publication.

Best regards,

Sarolta Tóth & Agnes Enyedi
Corresponding authors
Transfusion Medicine
Semmelweis University
Budapest, Hungary